# Hexokinase 2 is an RNA-binding protein that regulates mRNA translation independently of glycolysis and induces melanoma cell proliferation

Ana Luisa Dian[1,2,3], Lucilla Fabbri[1,2,3☯], Antoine Moya-Plana[4,5,6☯], Giuseppina Claps[4,5☯], Juliana C. Ferreira[7], Céline M. Labbé[1,2,3], Virginie Quidville[4,5], Sylvain Martineau[1,2,3], Dorothée Baille[1,2,3], Laetitia Besse[8], Cédric Messaoudi[8], Séverine Roy[4,5], Virginie Raynal[9], Sylvain Baulande[9], Wael M. Rabeh[7], Caroline Robert[4,5], Stéphan Vagner[1,2,3]*

**1** Institut Curie, PSL Research University, CNRS UMR 3348, INSERM U1278, Orsay, France, **2** Université Paris-Saclay, CNRS UMR 3348, INSERM U1278, Orsay, France, **3** Equipe labellisée Ligue contre le Cancer, Orsay, France, **4** INSERM U.981, Gustave Roussy, Villejuif, France, **5** Université Paris Sud, Université Paris-Saclay, Kremlin-Bicêtre, France, **6** Head and Neck Oncology Department, Gustave Roussy, Villlejuif, France, **7** Science Division, New York University Abu Dhabi, Abu Dhabi, United Arab Emirates, **8** Institut Curie, Université P.S.L., CNRS UAR2016, Inserm U.S.43, Université Paris-Saclay, Multimodal Imaging Center, Orsay, France, **9** Institut Curie Genomics of Excellence (ICGex) Platform, PSL Research University, Institut Curie Research Center, Paris, France

☯ These authors contributed equally to the work.
* stephan.vagner@curie.fr

## Abstract

Although metabolic benefits of glycolysis have been extensively described in tumor cells, the extra-metabolic functions linked to this energetic pathway in tumor growth and cell proliferation have not been clearly established yet. Recently, some key glycolytic enzymes, such as glyceraldehyde-3-phosphate dehydrogenase and pyruvate kinase 2, were reported to regulate mRNA translation. Translational control of gene expression is considered as a critical effector in cancer biology, representing a highly promising area of research. Here, we report that Hexokinase 2 (HK2), a glucose kinase that catalyzes the first step of glycolysis at the outer mitochondrial membrane (OMM), is an RNA-binding protein (RBP) that regulates mRNA translation in melanoma cell lines. Polysome profiling experiments followed by RNA sequencing indicate that the translational regulation exerted by HK2 is partly independent of the metabolic status or the glycolytic pathway. We found that HK2 specifically regulates translation of the mRNA encoding SOX10, a transcription factor implicated in the regulation of tumor initiation, maintenance, and progression in melanoma. RNA-protein interaction assays, including CrossLinking ImmunoPrecipitation (CLIP), indicate that HK2 is an RBP whose interaction with RNA is independent of its enzymatic activity, its ability to bind glucose or its association with the OMM. HK2 directly interacts with the 5′ untranslated region (5′UTR) of the *SOX10* mRNA through a stem-loop RNA secondary structure. Using RNA-protein proximity ligation assays and a fluorescence-based

**Data availability statement:** All relevant data are within the paper and its Supporting information files. The datasets generated in this study have been deposited in the Gene Expression Omnibus repository (GEO) under the accession number GSE274146.

**Funding:** This work was supported by Institut Curie (to SV), Gustave Roussy (to CR), Centre National de la Recherche Scientifique (CNRS) (to SV), Institut National de la Santé et de la Recherche Médicale (Inserm) (to SV and CR) and Agence Nationale de la Recherche (ANR) (ANR-10-EQPX-03 and ANR-10-INBS-09-08). Horizon Europe Marie Sklodowska-Curie Actions (MSCA) and FRM (Fondation pour la Recherche Medicale) provided the PhD fellowship to ALD. Marie Sklodowska-Curie Campus France Fellowship Prestige-2017-3-0017 provided the post-doctoral fellowship to GC. The funders had no role in study design, data collection and analysis, decision to publish, or preparation of the manuscript.

**Competing interests:** The authors have declared that no competing interests exist.

**Abbreviations:** CLIP, CrossLinking ImmunoPrecipitation; ECAR, Extra-cellular acidification rate; EMSA, Electrophoretic mobility shift assays; EMT, Epithelial–mesenchymal transition; FL, Full length; GAPDH, Glyceraldehyde-3-phosphate dehydrogenase; GTEx, Genotype-Tissue Expression; HK2, Hexokinase 2; MBD, Mitochondrial-binding deficient; OCR, Oxygen consumption rate; OMM, Outer mitochondrial membrane; ORF, Open reading frame; OXPHOS, Oxidative phosphorylation; PCA, Principal components analysis; PKM2, Pyruvate kinase 2; PLA, Proximity ligation assay; PNK, Polynucleotide kinase; RBP, RNA-binding protein; RIBOmap, Ribosome-bound mRNA mapping; RIP, RNA immunoprecipitation; SRY, Sex Determining Region Y; STR, Short tandem repeat; TCGA, The Cancer Genome Atlas; VST, Variance stabilizing transformation; 2C, Complex Capture; 5′UTR, 5′ untranslated region.

ribosome-bound mRNA mapping method, we found that high glucose conditions, which promote the release of HK2 from the OMM, induce an increase in HK2-*SOX10* mRNA interaction and *SOX10* mRNA translation in the cytoplasm. We further showed that HK2-dependent *SOX10* mRNA translation is involved in melanoma cell proliferation and colony formation. Collectively, our data highlight a nonmetabolic function of HK2 acting as an RBP and translation regulator.

## Introduction

Aerobic glycolysis, in which glucose is converted into lactate, is a key metabolic pathway in tumorigenesis. This pathway is usually promoted by cancer cells even when local conditions are suitable to utilize the oxidative phosphorylation (OXPHOS), a more efficient mechanism to generate ATP. This specific metabolic process is called the Warburg effect [1,2]. Although metabolic benefits of glycolysis have been established for sustaining biosynthesis and cellular proliferation in tumor cells [3,4] some data suggest that extra-metabolic functions are linked to this energetic pathway.

Over the past decades, several enzymes involved in the intermediary metabolism were shown to directly interact with RNAs [5]. Indeed, several metabolic enzymes were found to bind RNA in the vicinity of their substrate-binding pockets [6,7]. One example is the Rossmann fold, a di-nucleotide domain typically associated with the binding of nucleotides such as the nicotinamide adenine dinucleotide [8]. This fold plays a crucial role in the function of many enzymes, particularly dehydrogenases such as glyceraldehyde-3-phosphate dehydrogenase (GAPDH), which catalyze oxidation-reduction reactions. Interestingly, this same domain has been implicated in mediating RNA-binding [7]. Beyond dehydrogenases, several other enzymatic activities are found in the glycolytic pathway, including kinases for glucose (HK1/2), pyruvates (PKM1/2), and enolases (ENO1/2/3). While pyruvate kinase 2 (PKM2) and ENO1 (alpha-enolase) have been found to bind RNAs, the possible RNA-binding activity of hexokinase 2 (HK2), the first and limiting glycolysis enzyme catalyzing the irreversible phosphorylation of glucose to form glucose-6-phosphate, has never been addressed.

High levels of HK2 expression have been reported in many different types of cancer, and this upregulation is typically associated with poor outcomes in patients [9–13]. Although the role of HK2 in tumorigenesis has been attributed to its glycolytic activity, HK2 has also been shown to execute noncanonical functions that often regulate processes that are highly relevant for cell transformation and cancer development. In particular, Blaha and colleagues (2022) demonstrated a novel noncatalytic mechanism by which HK2 contributes to epithelial–mesenchymal transition (EMT) and metastasis in breast cancer via the transcription factor SNAIL, a key EMT inducer [14]. Intriguingly, HK2 has also been identified as a candidate RNA-binding protein (RBP) by orthogonal large-scale quantitative methods [15–18] and many glucose-binding proteins were recently reported to bind RNA in human cell lines [19]. Together with recent evidences showing that some key-glycolytic enzymes, such

as GAPDH and PKM were able to regulate the translation of mRNAs in human T-cells and mouse embryonic stem cells, respectively [20,21], we explored the role of HK2 as an RBP regulating mRNA translation in human cancer cells.

Here, we describe a novel extra-glycolytic function of HK2 in melanoma cell lines chosen due to the critical role of glycolysis and OXPHOS in melanoma development [22,23]. We demonstrate that the HK2 glucose kinase is a *bona-fide* RBP that regulates the translation of the *SOX10* mRNA, thereby regulating cancer-relevant phenotypes.

## Results

### HK2 regulates mRNA translation independently of the metabolic status of the cells

Upon stress stimuli and/or acquisition of mutations, melanocytes evolve from early superficial melanoma with radial growth (RGP) to early invasive melanoma (VGP) and metastatic tumor. We evaluated the level of HK2 in a variety of melanoma cell lines. We observed increased levels of HK2 in invasive cell lines (A375, WM35, SKMel10) compared to early invasive melanoma (VGP: WM793), early superficial melanoma with radial growth (RGP: SBCL2) or normal human melanocytes (S1A Fig). In addition, in patients with cutaneous melanoma, HK2 expression was significantly higher in stages III–IV compared to stages I–II ($p = 0.026$) (S1B Fig).

Since PKM2 and GAPDH were shown to associate with polysomes [21,24], we analyzed whether HK2 could also be associated with polysomes. To this end, we performed western blot analysis on polysome fractions isolated from A375 cells, one of the melanoma cell lines with the highest level of HK2 (S1C Fig). We found that, as for PKM2 and GAPDH, HK2 is present in ribosome-containing fractions (Fig 1A). This is specific since another glycolytic enzyme, LDHB (lactate dehydrogenase B), is not associated with ribosome-containing fractions. This ribosome-specific association was validated by treating lysates of A375 cells with EDTA prior to sucrose gradient fractionation. Upon EDTA-induced polysome disassembly, HK2 shifted from polysome fractions to lighter fractions, similarly to the ribosomal protein S6 (Fig 1B).

To investigate whether HK2 is involved in the translational regulation of specific mRNAs, we depleted (siRNA) HK2 in A375 melanoma cells. Of note, depletion of HK2 did not lead to a decrease in the levels of GAPDH and PKM2 (S2A Fig). We used RNA sequencing to simultaneously quantify the abundance of total transcripts as well as those that are being actively translated (translatome), *i.e.,* that are associated with polysomes (>4 ribosomes). The translatome data were normalized to the expression levels of the corresponding mRNAs. The analysis of mRNAs that are differentially recruited to heavy polysome fractions in HK2-depleted cells showed that the translation status of 101 mRNAs was changed (log2(fold change) > 0.7; *p*-value < 0.05) (Fig 1C and S1 Table).

Translation regulation by HK2 could be the consequence of altered metabolism since siRNA-mediated depletion of HK2 results in decreased lactate secretion (S2B Fig), indicating a reduced glycolytic activity. To determine whether this translational regulation is linked to the metabolic status of the cells, we cultured A375 in medium containing galactose instead of glucose. The metabolism of galactose eventually converges with the glucose metabolism through the Leloir pathway. However, because this conversion occurs at a much slower rate than glucose entry into glycolysis, culture with galactose favors OXPHOS [25–27]. To confirm that A375 switch their metabolic profile in the presence of galactose, we employed the Seahorse metabolic flux assay to measure the oxygen consumption rate (OCR) and the extra-cellular acidification rate (ECAR) of living cells in culture. We observed that the ECAR, an indicator of aerobic glycolysis, was significantly higher in A375 cells cultured in glucose-containing medium compared to cells cultured in galactose. On the other hand, the OCR, an indicator of OXPHOS, was higher than ECAR in cells cultured in galactose-containing medium (S2C Fig). These results therefore confirmed the metabolic switch of melanoma cells from glycolysis to mitochondrial respiration. We found that modifying the metabolic status of cells by growing them in the presence of galactose instead of glucose impacted the translation of 312 mRNAs (Fig 1D and S1 Table). Only about 20% of mRNAs regulated by HK2 are also regulated by galactose, indicating that translation regulation by HK2 is partly independent of the metabolic status of the cells (Fig 1D). We also compared the subsets of mRNAs translationally regulated by HK2, PKM2, and GAPDH. We found

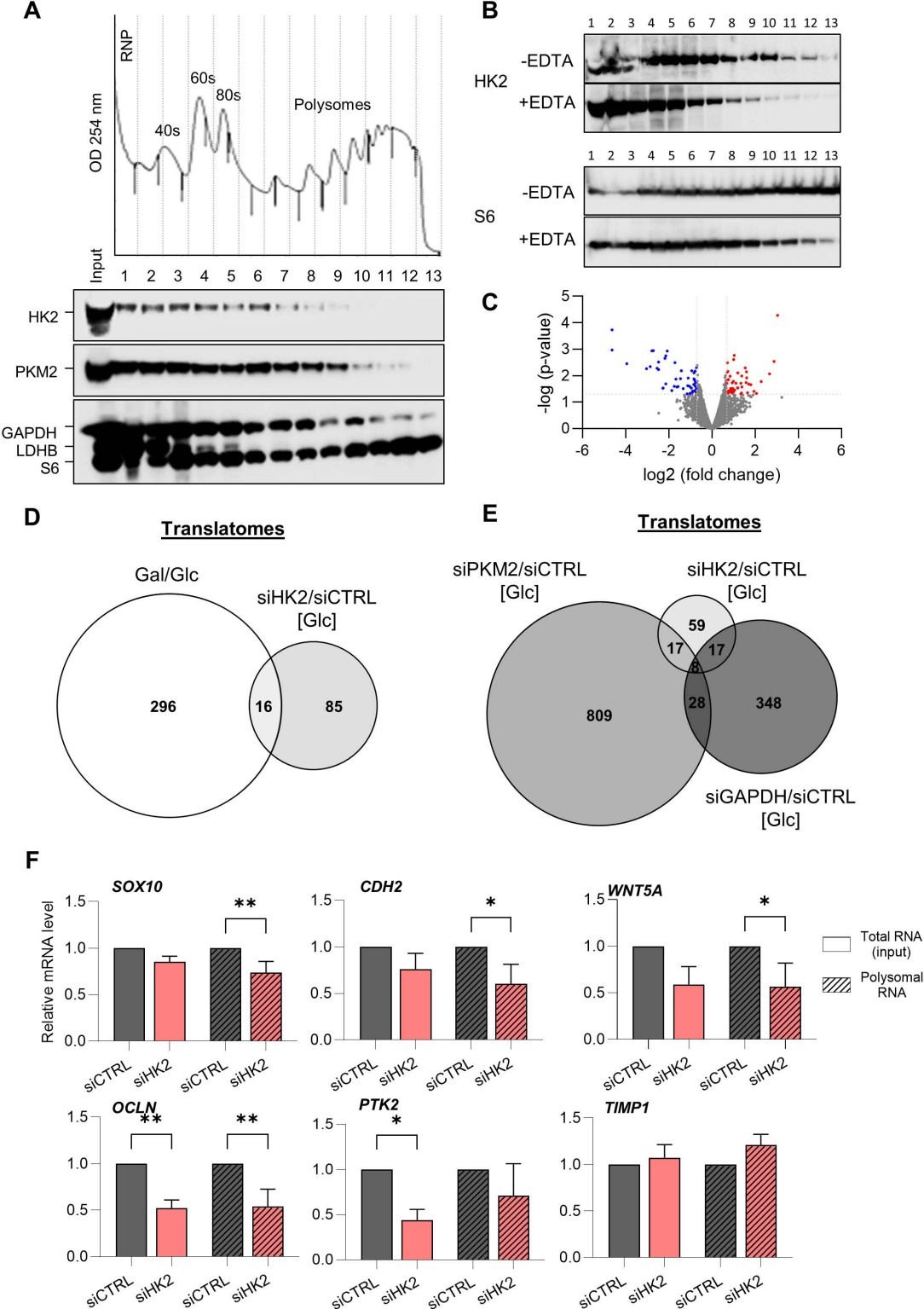

**Fig 1. HK2 regulates mRNA translation in melanoma cells.** **(A)** Western blot analysis of HK2, PKM2, GAPDH, LDHB, and the ribosomal protein S6 in fractions (horizontal axes) obtained by sucrose-gradient (10%–50%) ultracentrifugation of lysates from A375 cells. **(B)** Western blot analysis of HK2 and the ribosomal protein S6 in fractions (horizontal axes) obtained by sucrose-gradient (10%–50%) ultracentrifugation of EDTA-treated lysates from

A375 cells. **(C)** Volcano plot representing translationally deregulated mRNAs in A375 melanoma cells upon siRNA-mediated depletion of HK2 (siHK2) in comparison to control (siCTRL). Red dots represent mRNAs translationally upregulated upon HK2 depletion, while blue dots represent mRNAs translationally downregulated. X-axis denotes log2 fold change values, and Y-axis denotes −log10 $p$-values. **(D)** Comparison of translatomes between A375 cells cultured in galactose-containing medium relative to glucose (312 deregulated mRNAs, 214 being upregulated and 98 downregulated) and siRNA-mediated depletion of HK2 in cells cultured in glucose-containing medium relative to control (siHK2/siCTRL; 101 deregulated mRNAs, being 52 upregulated and 49 downregulated) ($n$ = 3 biological replicates). $P$-value ≤0.05 with fold change >0.7 (DESeq2). **(E)** Comparison of translatomes between A375 cells upon siRNA-mediated depletion of key glycolytic enzymes (HK2, GAPDH, PKM2) in A375 cells relative to control. 401 mRNA candidates (269 upregulated and 132 downregulated) were found deregulated upon siRNA-mediated depletion of GAPDH relative to control (siGAPDH/siCTRL), and 862 mRNA candidates (554 upregulated and 308 downregulated) were found deregulated upon siRNA-mediated depletion of PKM2 (siPKM2/siCTRL) ($n$ = 3 biological replicates). $P$-value ≤0.05 with fold change >0.7 (DESeq2). **(F)** RT-qPCR analysis of gene expression upon HK2 depletion. Transcriptional (total RNA; without black stripes) and translational (polysomal RNA; with black stripes) levels of each gene were obtained from polysome profiling of A375 cells transfected with siRNAs targeting HK2 (siHK2, light red) or control (siCTR, gray). p-values were calculated by ordinary two-way ANOVA with Dunnett's multiple comparisons test (SD, $n$ = 3 biological replicates) and only significant comparisons are shown (* $p$ ≤ 0.05; ** $p$ ≤ 0.01). HK2 depletion significantly decreased the abundance of *SOX10*, *CDH2*, and *WNT5A* mRNAs extracted from polysome fractions, but not in total RNAs. Of note, these mRNAs were not found in the translatome data as regulated by HK2 because of a lack of statistical significance (*SOX10*, $p$ = 0.3; *OCLN*, $p$ = 0.4; *WNT5A*, $p$ = 0.7). The individual numerical values for panels Fig 1C, 1F are available at S1 Data.

that the translation of 59 mRNAs is specifically regulated by HK2, but not by the other tested glycolytic enzymes (i.e., PKM2 or GAPDH) (Fig 1E), indicating that HK2-mediated translational regulation is not a consequence of altered cellular metabolism.

### HK2 regulates the translation of the *SOX10* mRNA encoding the Sex Determining Region Y (SRY)-Box Transcription Factor

In parallel with our sequencing-based approach that did not identify genes with known functions in cancer (S1 Table), we conducted a targeted RT-qPCR-based small-scale screen using the RT$^2$ Profiler PCR Array (Qiagen, #PAHS-090Z), which includes a panel of 84 cancer-associated mRNAs (S2 Table) encoding proteins involved in key oncogenic processes such as cell differentiation and development, proliferation, migration, and transcriptional regulation. We identified 6 mRNA candidates that were less associated with polysomes (>4 ribosomes) upon siRNA-mediated HK2 depletion (S3A Fig) and were not found in the sequencing-based approach. Validation of the results from this screen was done by analyzing the abundance of the 6 mRNA candidates in polysome fractions isolated from HK2-depleted A375 cells, as well as in total RNAs. Depletion of HK2 led to a significant decrease in the abundance of 3 of them (*SOX10, CDH2, and WNT5A*) in RNAs extracted from polysome fractions, but not in total RNAs (Fig 1F). Of note, these mRNAs were not found in the list of 101 candidate mRNAs regulated at the translation level by HK2 (S1 Table) because of a lack of statistical significance ($p$ > 0.05).

Among the three validated mRNAs, we were particularly interested in the *SOX10* mRNA that encodes the Sex Determining Region Y (SRY)-Box Transcription Factor 10 (SOX10), which has a well-known activity as a transcription factor regulating tumor initiation, maintenance, and progression in melanoma [28–30]. Also, when comparing SOX10 expression in different types of malignancies in the TCGA database, we found that cutaneous melanoma highly expressed SOX10 compared to other cancers (S4A Fig). Moreover, in a cohort of melanoma patients, the quartile low-SOX10 group had a significantly higher overall survival than the quartile high-SOX10 group ($p$ = 0.012) (S4B Fig). Interestingly, when the patients were stratified according to their mutational profile, SOX10 expression appeared to be a powerful prognostic factor of oncologic outcomes in patients with BRAF-mutated melanoma (as A375 cell line) while no prognostic value was observed for RAS, NF1, and triple-negative melanomas (S4C Fig). This could be explained by the fact that aerobic glycolysis is upregulated in human BRAF$^{V600}$ melanoma cells via transcriptional regulation of HK2 and GLUT1 [31]. Thus, the transcriptional activation of HK2 by the MAPK/ERK pathway may subsequently lead to higher expression of SOX10.

To confirm the *SOX10* mRNA translational regulation by HK2, we monitored the level of *SOX10* mRNA isolated from all polysome fractions of A375 cells upon siRNA-mediated depletion of HK2 (S3B Fig). We found that *SOX10* mRNA shifted

from heavy polysome fractions to lighter polysome fractions upon HK2 depletion, indicating that HK2 depletion decreases the translation of the *SOX10* mRNA (Fig 2A). Consistently, HK2 depletion decreased SOX10 protein levels, but not *SOX10* mRNA levels, in four different BRAF-mutated (V600E) melanoma cell lines (Figs 2B–2D and S3C). This regulation appears to be specific to HK2 since the depletion of GAPDH did not affect SOX10 expression. In addition, the effect of HK2 depletion on SOX10 expression could also be confirmed using a short hairpin RNA (shHK2) (Figs 2E and S3F, S3G). Collectively, our results demonstrate that HK2 regulates the translation of the *SOX10* mRNA.

To define the *cis*-acting elements involved in the translational regulation of the *SOX10* mRNA, we used reporter plasmids producing mRNAs containing the 5′ and 3′UTR of *SOX10* mRNA either upstream or downstream of a Renilla luciferase (Rluc) ORF, respectively. We also generated six *SOX10* 5′UTR deletion mutants containing distinct deletions of ~50 nucleotides each (Δ1–6) (Fig 2F). We transiently transfected the reporter plasmids together with a control Firefly luciferase (Fluc) expressing plasmid into A375 and A2058 cells and quantified the relative Renilla over Firefly luciferase activities (Rluc/Fluc ratio). We observed a higher Rluc/Fluc ratio for the construct containing the 5′UTR compared to the 3′UTR or no UTR ("empty"), showing the importance of the 5′UTR in the translation of the *SOX10* mRNA (Fig 2G and 2H). We also observed that the constructs expressing the Δ4, Δ5, and Δ6 deletions, but not the Δ1, Δ2, and Δ3 deletions, presented a reduced Rluc/Fluc ratio when compared to the 5′UTR-containing plasmid, with a significant effect for the Δ4 deletion (Fig 2G and 2H). Since the Rluc/Fluc activity is presented as normalized to the RNA expression levels of the different constructs, we concluded that the Δ4 region is essential for efficient translation and not mRNA stability.

## HK2 directly binds RNA

To test the hypothesis that HK2 directly interacts with RNA, we first performed Complex Capture (2C) experiments [32]. This approach takes advantage of the ability of negatively charged nucleic acids to bind silica matrix-based columns, consequently retaining UV-crosslinked RNA-RBP complexes. HK2 was retained on the column in extracts of UV-C irradiated cells, but not detected in RNase A-treated samples, demonstrating that HK2 was retained on the column due to its UV-C-induced covalent interaction with RNAs (Fig 3A). The well-characterized RBP HuR was used as a positive control in this assay, as well as the noncanonical RNA-metabolic enzymes GAPDH and PKM2. The metabolic enzymes ACOT1/2 and HSC70 were used as negative controls.

To gain more evidence that HK2 binds to RNA, we performed CrossLinking ImmunoPrecipitation (CLIP) assay [33,34]. This assay relies on the UV-C-dependent crosslinking of RNA-RBP complexes in living cells followed by cell lysis and partial RNA digestion with RNase I. The RNA molecules crosslinked to HK2 were then recovered through HK2 immunoprecipitation and radioactively labeled with $^{32}$P-yATP and T4 polynucleotide kinase (PNK). We observed a radioactive signal at the molecular size of HK2 in UV-C-treated samples (Figs 3B and S5A). The signal was not detected in the noncrosslinked conditions or in IgG-immunoprecipitated conditions (negative controls). The signal was modulated according to the RNase I concentration used in UV-C crosslinked cells; at higher concentrations, the radioactive signal collapsed to a sharp band at the molecular mass of HK2. In addition, we sought to confirm the nature of the nucleic acids bound by HK2. To this end, we extracted the radioactively ($^{32}$P-yATP) labeled nucleic acids from the nucleic acid-HK2 complexes. The purified nucleic acids were then migrated on a denaturing TBE-urea gel, and the radioactive signal observed by autoradiography. The nucleic acids bound to HK2 migrated as a smear above 100 nts (S5B Fig). The treatment of A375 cell lysates with increasing concentrations of RNase I modulated the size of the observed smear until it collapsed below 30−20 nts, thus indicating that HK2 interacts with RNAs. To further validate the direct and specific binding of HK2 to RNA, we performed CLIP assays in melanoma cells following siRNA-mediated HK2 depletion. The radioactive signal observed as a sharp band at the molecular mass of HK2 was reduced upon HK2 depletion (S5C and S5D Fig), thereby confirming the specificity of the signal as HK2-RNA complexes. Taken together, these results indicate that HK2 directly interacts with RNA in living melanoma cells.

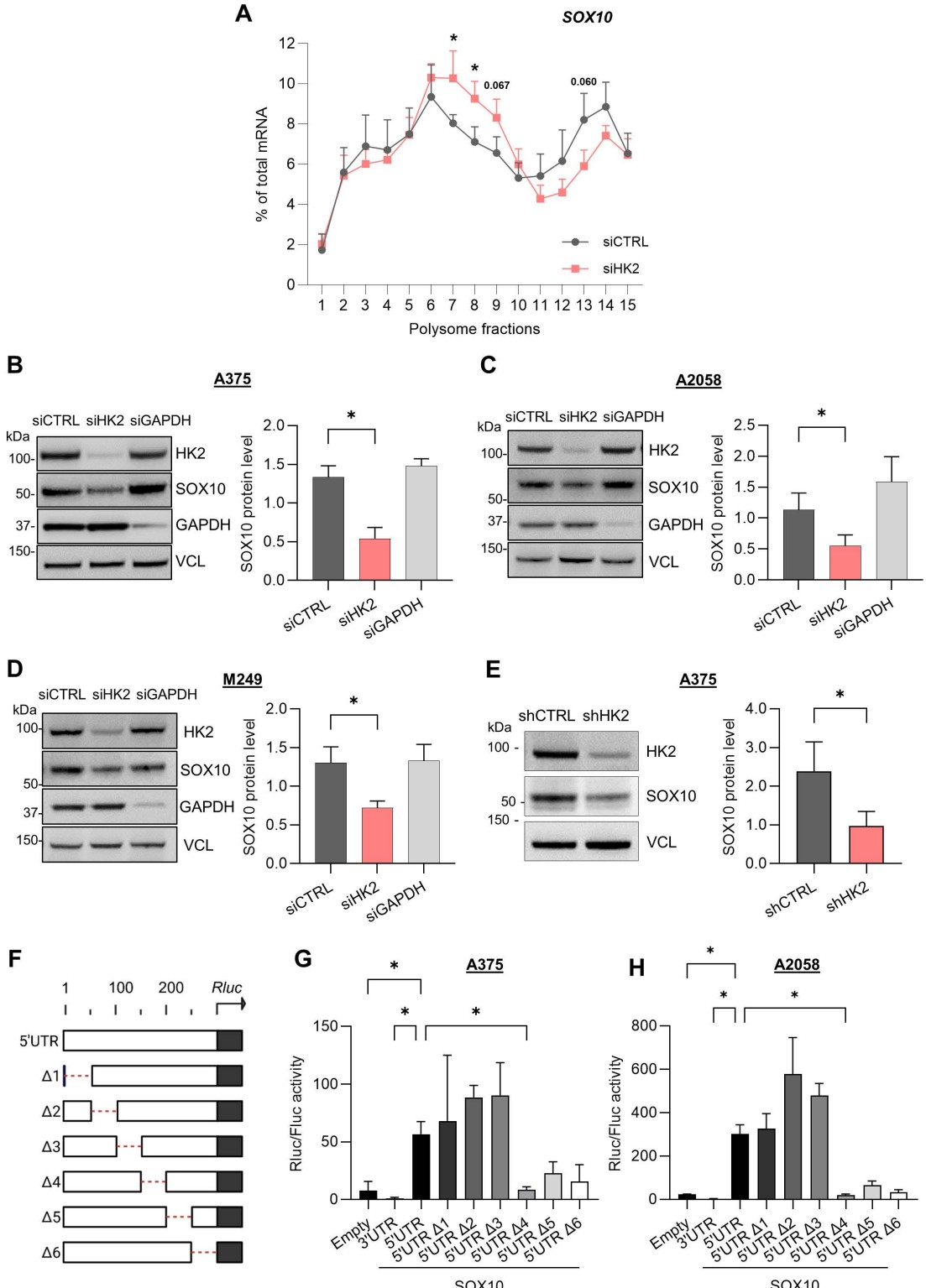

**Fig 2. HK2 regulates the translation of the *SOX10* mRNA. (A)** RT-qPCR quantification representing the % of *SOX10* mRNA in fractions (horizontal axes) obtained by sucrose-gradient (10%–50%) ultracentrifugation of lysates from A375 cells transfected with siRNAs targeting HK2 (siHK2, light red) or control (siCTRL, gray). The *SOX10* mRNA is normalized to *ACT* mRNA. *p*-values were calculated by unpaired, two-tailed Student *t* test (SD, *n* = 3

biological replicates) and significant comparisons are shown (* $p \leq 0.05$). **(B, C, and D)** Western blot analysis of the SOX10 protein level in distinct melanoma cell lines transfected with siRNAs targeting HK2 (siHK2, light red), GAPDH (siGAPDH, light gray), or control (siCTRL, gray). The SOX10 protein quantification is normalized to VCL expression. $p$-values were calculated by two-tailed unpaired $t$ test (SD, n = 3 biological replicates) and only significant comparisons are shown (* $p \leq 0.05$). **(E)** Western blot analysis of the SOX10 protein level in A375 cells upon stable HK2 knockdown. The SOX10 protein quantification is normalized to VCL expression. $p$-value was calculated by two-tailed unpaired $t$ test (SD, $n = 3$ biological replicates) (* $p \leq 0.05$). **(F)** Schematic of full-length (FL) and mutated SOX10 5′UTR (Δ1–6) reporters cloned upstream of the Renilla (RLuc) ORF. The deleted regions of ~50nts are represented as red dashed lines. **(G)** Luciferase assay performed in A375 cells co-transfected with a Firefly reporter and the RLuc reporters in (G). An empty RLuc vector and the SOX10 3′UTR RLuc reporter were used as controls. The activity of the Firefly and RLuc reporters was measured 48 h after transfection. The RLuc activity was normalized by the Firefly activity, and the data shown is relative to the Empty vector. The luciferase data is presented as normalized to the RNA expression levels of the different vectors. $p$-values were calculated by ordinary two-way ANOVA with Dunnett's multiple comparisons test (SD, $n = 4$) and only significant comparisons are shown (* $p \leq 0.05$). **(H)** Luciferase assay performed in A2058 cells in the condition described in H. $p$-values were calculated by ordinary ANOVA with Dunnett's multiple comparisons test (SD, $n = 3$) and only significant comparisons are shown (* $p \leq 0.05$). The individual numerical values for panels Fig. 2A–2E, 2G, 2H are available at S2 Data.

We next examined whether the RNA-binding activity of HK2 was dependent on its hexokinase activity or subcellular localization. We ectopically expressed GFP-tagged HK2, either full length (HK2-FL) or mutants, in HEK293T cells and performed the CLIP assay. The first mutant used is the HK2-MBD mutant, which carries a deletion of the first 15 amino acids of the protein, which are critical for the association of HK2 with the OMM through its interaction with the voltage-dependent anion-selective channel 1 (VDAC-1) [35]. This binding has been shown to enhance glycolysis due to privileged access of HK2 to newly synthesized ATP generated by the mitochondria. However, HK2 has been shown to alternate between cytoplasmic and mitochondrial-bound states in response to environmental and metabolic stress [36]. The second tested mutant was the HK2-DA mutant, in which alanine is substituted to two aspartic acid residues in both the amino- and carboxy-terminal domains of HK2 (D209A/D657A). Mutations in both sites inhibit HK2 binding to glucose [37]. Additionally, we tested whether the catalytic activity of HK2 is necessary for its binding to RNA. To this end, alanine was substituted for two serine residues in both the amino- and carboxy-terminal domains of HK2 (HK2-SA mutant; S155A/S603A) [36]. As expected, we observed a sharp radioactive signal at the molecular size of HK2-FL-GFP (~129 kDa), which corresponds to the upper band (Figs 3C and S5E). The lower band, labeled with an asterisk, corresponds to a nonspecific signal. Of note, no radioactive signal was detected in GFP-only-transfected cells. Therefore, all mutants retained their ability to bind RNA, indicating that the RNA-binding activity of HK2 does not rely on its enzymatic activity, its ability to bind glucose or its association with the OMM.

To further study HK2-RNA interaction, we used the R-Deep method [38]. RNase-treated or untreated lysates of A375 cells were loaded onto sucrose density gradient (5%–25%), ultracentrifuged, and fractionated. We founds that 18% of the total cellular HK2 changed its distribution in the sucrose gradient following RNase treatment, indicating that the composition of HK2-containing complexes is partially dependent on RNA (Fig 3D). Intriguingly, complexes containing either GAPDH or PKM2 showed less sensitivity to RNase treatment (S6 Fig), therefore suggesting that HK2-containing complexes exhibit a greater dependence on RNA than complexes containing the other known RNA-binding glycolytic enzymes. In contrast to HuR, HK2 presented an RNA-dependent shift to more dense fractions suggesting that the presence of RNA might prevent the formation of heavy HK2-containing complexes. Proteins exhibiting shifts in their distribution to denser fractions upon RNAse treatment have already been described in the R-Deep method [38]. Indeed, in contrast to the proteins found to shift to less dense fractions (i.e., HuR), dense-shifted proteins have no significant enrichment for domains linked to RNA-binding nor "RNA binding" molecular function. Upon RNA loss, these proteins are proposed to establish novel interactions potentially due to the reduction of repulsive RNA charges or clearance of binding sites.

## HK2 associates with the *SOX10* mRNA

To determine whether HK2 could specifically associate with the *SOX10* mRNA, we performed HK2 immunoprecipitation (IP) followed by RT-qPCR (RNA ImmunoPrecipitation-RIP analysis) in A375 and A2058 melanoma cells. We identified

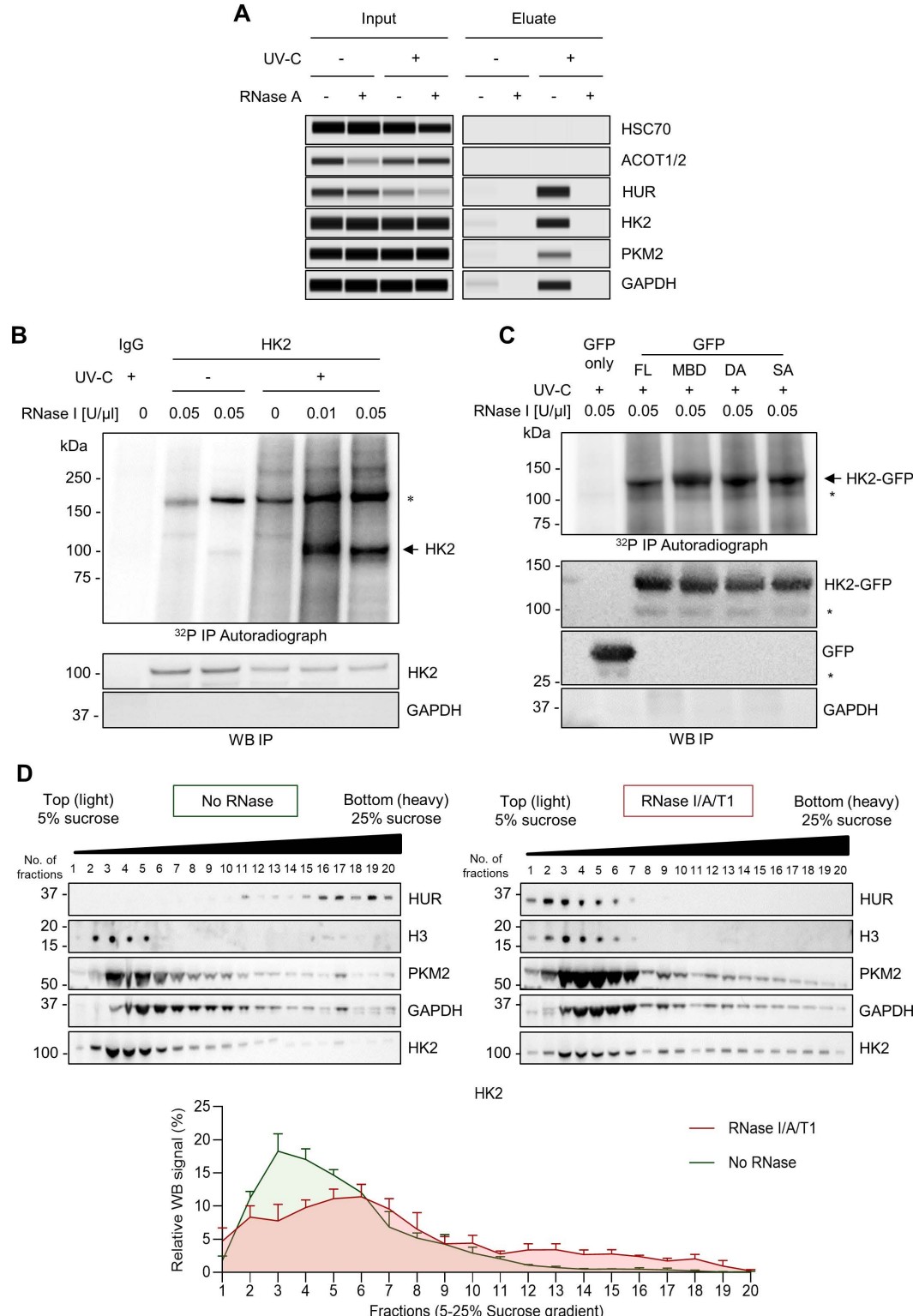

**Fig 3. HK2 binds directly to RNA *in vivo*. (A)** RNA-dependent retention of HK2 in silica matrix columns assessed by automated capillary western blot (ProteinSimple). Specific HK2 signal was detected around the expected molecular mass in lysates of UV-C irradiated A375 cells not treated with RNase A prior to loading to the silica column. HUR, GAPDH, and PKM2 were used as positive controls, and HSC70 and ACOT1/2 as negative controls.

**(B)** CLIP from endogenous HK2 in A375 cells (*n* = 3 biological replicates). Upper panel: Autoradiography of HK2-RNA complexes in UV-C-treated (+) or untreated (−) A375 cells. One major band is observed in the expected molecular mass of HK2, as indicated by a black arrow, and the asterisk (*) indicates a nonspecific band. Lower panel: western blots of HK2 and IgG immunoprecipitation. Same conditions as the upper panel. **(C)** CLIP from GFP-only (control) or GFP-HK2-transfected HEK293T cells (*n* = 3 biological replicates). Upper panel: autoradiography of HK2-GFP-RNA complexes in UV-C-treated (+) cells. Lower panel: western blots of GFP immunoprecipitation. Same conditions as the upper panel. The arrow indicates the expected HK2-GFP molecular mass, and the asterisk (*) indicates a nonspecific band. FL, full length; MBD, mitochondrial-binding deficient; DA, nonglucose-binding mutant; SA, catalytically inactive mutant. **(D)** Sucrose density gradient (5%–25%) centrifugation and fractionation of A375 cell lysates treated with RNase I/A/T1 or left untreated (SD, *n* = 4 biological replicates). Upper panel: western blot of the sucrose-gradient fractions (horizontal axes) obtained after ultracentrifugation of lysates. HUR was used as a positive control, and H3 as a negative control. Lower panel: western blot quantification representing the % of HK2 in each sucrose fraction. The individual numerical values for panels Fig 3D are available at S3 Data.

an enrichment of the *SOX10* mRNA in HK2 IP over the control IP using IgG antibodies of the same isotype (Figs 4A, 4B and S7A, S7B). mRNAs of similar or higher abundance, such as *GAPDH*, *TBP,* and *ACT* were less enriched in the HK2 IP, indicating that HK2 specifically binds to the *SOX10* mRNA. To delineate the HK2 binding site on the *SOX10* mRNA, *CatRAPID* fragment prediction [39] was employed and revealed a significant binding region to the *SOX10* sequence located between nucleotides 206 and 321. This region corresponds to the 5′ untranslated region (5′UTR) of the *SOX10* mRNA (S7C Fig). We transfected A375 cells with plasmids expressing the RLuc ORF containing the *SOX10* 5′UTR, the 3′UTR, or no *SOX10* sequences (empty) and performed RIP experiments. Consistent with the *in silico* prediction, we observed a significant enrichment of the 5′UTR-Rluc mRNA over the Rluc-3′UTR mRNA (Figs 4C and S7D). We next investigated whether the 3′ half of the *SOX10* 5′UTR, found to be essential for its efficient translation, is also required for HK2 association. We found that the deletion of nts 152–201 (Δ4) in the *SOX10* 5′UTR significantly decreases the association between HK2 and the Rluc mRNA (Figs 4D and S7D). Together, these results indicate that the sequence located between nucleotides 152 and 201 in the *SOX10* 5′UTR is critical not only for mediating *SOX10* 5′UTR-driven translation but also for HK2 association.

To provide additional evidence that HK2 associates with the 5′UTR of the *SOX10* mRNA, we used an immunofluorescence-based RNA-Proximity Ligation Assay (RNA-PLA) [40] (Fig 4E) that enables the study of the association of proteins of interest with their target RNAs in situ. Here we used an antibody that specifically recognizes HK2, in combination with an antisense probe that hybridize adjacent to the predicted HK2-*SOX10* interaction region within the 5′UTR. To ensure the specificity of the PLA signal, we depleted *SOX10* (Fig 4F–4H) or *HK2* (Fig 4I–4K) using specific siRNAs. In control conditions, the association of HK2 with the *SOX10* mRNA was observed by a robust PLA signal. SiRNA-mediated depletion of HK2 or *SOX10* mRNA resulted in a loss of PLA signal compared to control siRNA, thereby confirming the specificity of the signal for HK2-*SOX10* 5′UTR interactions. Together, these results indicate that HK2 preferentially associates with *SOX10* mRNA via its 5′UTR.

## HK2 directly interacts with a stem-loop RNA secondary structure within the *SOX10* 5′UTR

We investigated the interaction and structure of the HK2-*SOX10* mRNA complex using *AlphaFold 3* [41]. The 152–201 nucleotide region within the 5′UTR of *SOX10*, previously suggested as essential for HK2–*SOX10* mRNA interactions, was used for this prediction. The RNA molecule (cyan/red in Fig 5A) adopts a helical secondary structure with a curved, loop region (194–198 nts) positioned near to a lysine-/arginine-rich region in the N-terminal domain of HK2 (green in Fig 5A). The red-highlighted region in Fig 5A represents a potential interaction interface between the *SOX10* mRNA and HK2.

To assess the direct and specific binding of HK2 to the predicted *SOX10* stem-loop structure, we performed *in vitro* electrophoretic mobility shift assays (EMSA). Based on the predicted RNA-binding site in the *SOX10* 5′UTR, a 30 nt-long synthetic *SOX10* RNA probe, fluorescently labeled at the 5′ end with ATTO700, was used (Fig 5B). Given that CLIP results suggested that the RNA-binding activity of HK2 is independent of its association with the OMM, we expressed and purified recombinant HK2 protein variants: HK2 Δ16 lacking its mitochondrial-binding peptide (deletion of the first 16

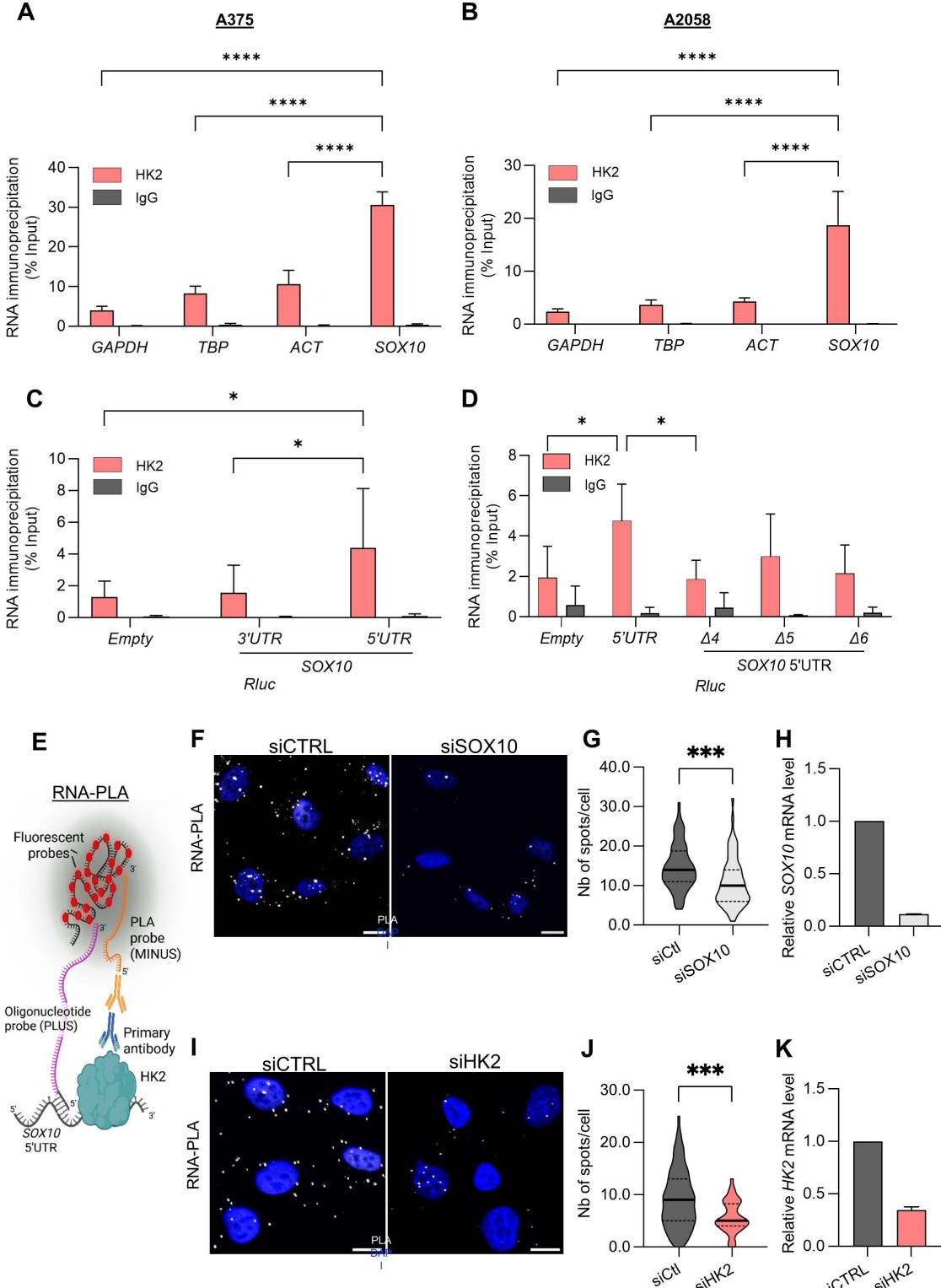

**Fig 4. HK2 associates with the 5′UTR of the *SOX10* mRNA.** RIP experiment performed on **(A)** A375 and **(B)** A2058 melanoma cell lines. *SOX10* mRNA was analyzed in IgG and HK2 immunoprecipitated samples and expressed as percentage of the mRNA present in the input. *GAPDH*, *TBP,* and *ACT* mRNAs were used as negative controls. *p*-values were calculated by ordinary two-way ANOVA with Dunnett's multiple comparisons test (SD, *n* = 3

biological replicates) (**** $p \le 0.0001$). **(C)** RIP experiment performed on A375 cells transfected with luciferase reporters containing the *SOX10* 5′ and 3′UTR sequences upstream of a *RLuc* reporter gene. An Empty reporter was used as control. *RLuc* mRNA was analyzed in IgG and HK2 immunoprecipitated samples and expressed as percentage of the mRNA present in the input. *p*-values were calculated by ordinary two-way ANOVA with Dunnett's multiple comparisons test (SD, $n = 6$ biological replicates) (* $p \le 0.05$). **(D)** RIP experiment performed on A375 cells transfected with RLuc luciferase reporters containing the *SOX10* 5′UTR FL and Δ4, Δ5, and Δ6. *RLuc* mRNA was analyzed in IgG and HK2 immunoprecipitated samples and expressed as percentage of the mRNA present in the input. *p*-values were calculated by ordinary two-way ANOVA with Dunnett's multiple comparisons test (SD, $n = 3$ biological replicates) (* $p \le 0.05$). **(E)** Schematic of the proximity ligation assay (PLA) used to detect HK2 protein interaction with SOX10 5′UTR. **(F)** Representative confocal images of *SOX10* mRNA-HK2 protein interaction detected by PLA (white spots) in A375 cells transfected with control siRNA (siCTRL) or with siRNA targeting SOX10 (siSOX10). Nuclei are stained with DAPI (blue). Scale bar: 10 μm. **(G)** Quantification of the PLA signal in the condition described in (F). The data shown represent the number of spots/cells from a representative experiment ($n = 2$ biological replicates). *p*-values were calculated by unpaired, two-tailed Student *t* test (*** $p \le 0.001$). **(H)** RT-qPCR quantification of *SOX10* mRNA level in the condition described in F. mean ± SD, $n = 2$ biological replicates. **(I)** Representative confocal images of *SOX10* mRNA–HK2 protein interaction detected by PLA (white spots) in A375 cells transfected with control siRNA (siCTRL) or with siRNA targeting HK2 (siHK2). Nuclei are stained with DAPI (blue). Scale bar: 10 μm. **(J)** Quantification of the PLA signal in the condition described in (I). The data shown represent the number of spots/cells from a representative experiment ($n = 2$ biological replicates). *p*-values were calculated by unpaired, two-tailed Student *t* test (*** $p \le 0.001$). **(K)** RT-qPCR quantification of *HK2* mRNA level in the condition described in d. mean ± SD, $n = 2$ biological replicates. The individual numerical values for panels Fig 4A–4D, 4G, 4H, 4J, 4K are available at S4 Data.

amino acids of its N-terminal domain), N-ter: N-terminal (residues 16–479) domain of the HK2 protein; C-ter: C-terminal (residues: 480–917) domain of the HK2 protein (Fig 5C). Increasing concentrations of HK2 Δ16 shifted the migration of the ATTO700-labeled *SOX10* SL RNA, resulting in slower-migrating species corresponding to HK2-RNA complexes (Fig 5D and 5E). In contrast, we observed that neither the separate N-terminal nor the C-terminal domains form complexes with the RNA (Fig 5D), indicating that the presence of both domains in the full-length HK2 enzyme is essential for binding the RNA. To further assess the specificity of the interaction, we employed competition experiments using nonfluorescently labeled competitor RNAs (*i.e.*, *SOX10* SL and mSL, mutated in the loop sequence, Fig 5F). We observed that increasing concentrations of unlabeled *SOX10* SL RNA almost completely prevented the interaction between HK2 Δ16 and the ATTO700-labeled *SOX10* SL RNA (Fig 5G and 5H). In contrast, the unlabeled *SOX10* mSL RNA only competes at significantly higher concentrations, requiring a 500- to 1000-fold molar excess to inhibit complex formation. These results strongly suggest that HK2 directly interacts with a specific stem-loop RNA secondary structure located in the 5′UTR of the SOX10 mRNA. Of note, we could not find a similar sequence motif in the 5′UTR of the other candidate HK2 translationally regulated mRNAs. Thus, HK2 directly interacts with a stem-loop RNA secondary structure located in the 5′UTR of the *SOX10* mRNA.

## HK2 release from the mitochondria regulates *SOX10* mRNA translation

Glucose, the substrate of HK2, is known to promote HK2 dissociation from mitochondria [36]. We therefore sought to explore whether HK2-mediated *SOX10* mRNA translation is regulated by glucose. We observed a significant decrease in the level of SOX10 protein when A375 cells were cultured at the lowest concentrations of glucose (Fig 6A). Supplementation of glucose-free media with the HK2 product G-6-P, which is known to release HK2 from mitochondria into the cytosol in a feedback mechanism [42] significantly increases SOX10 protein levels (Fig 6B). Since *SOX10* mRNA levels remained unchanged upon modulation of glucose levels (S8A Fig) or G-6-P supplementation (S8B Fig), these data suggest that SOX10 expression might be regulated at the translational level by glucose.

To address this, we used a fluorescence-based ribosome-bound mRNA mapping (RIBOmap) method [43] that enables the *in situ* detection of ribosome-bound mRNAs using three probes that, when in proximity, generate a DNA amplicon via rolling-circle amplification corresponding to active translation (Fig 6C). Actively translated mRNAs were visualized with a fluorescent probe complementary to the DNA amplicon. We detected RIBOmap signals in high glucose conditions (Fig 6D and 6E). The RIBOmap signals are specific since depletion of the *SOX10* mRNA resulted in loss of RIBOmap signals when compared to cells transfected with control siRNA (S9A–S9C Fig). Consistent with the variation in the SOX10 protein

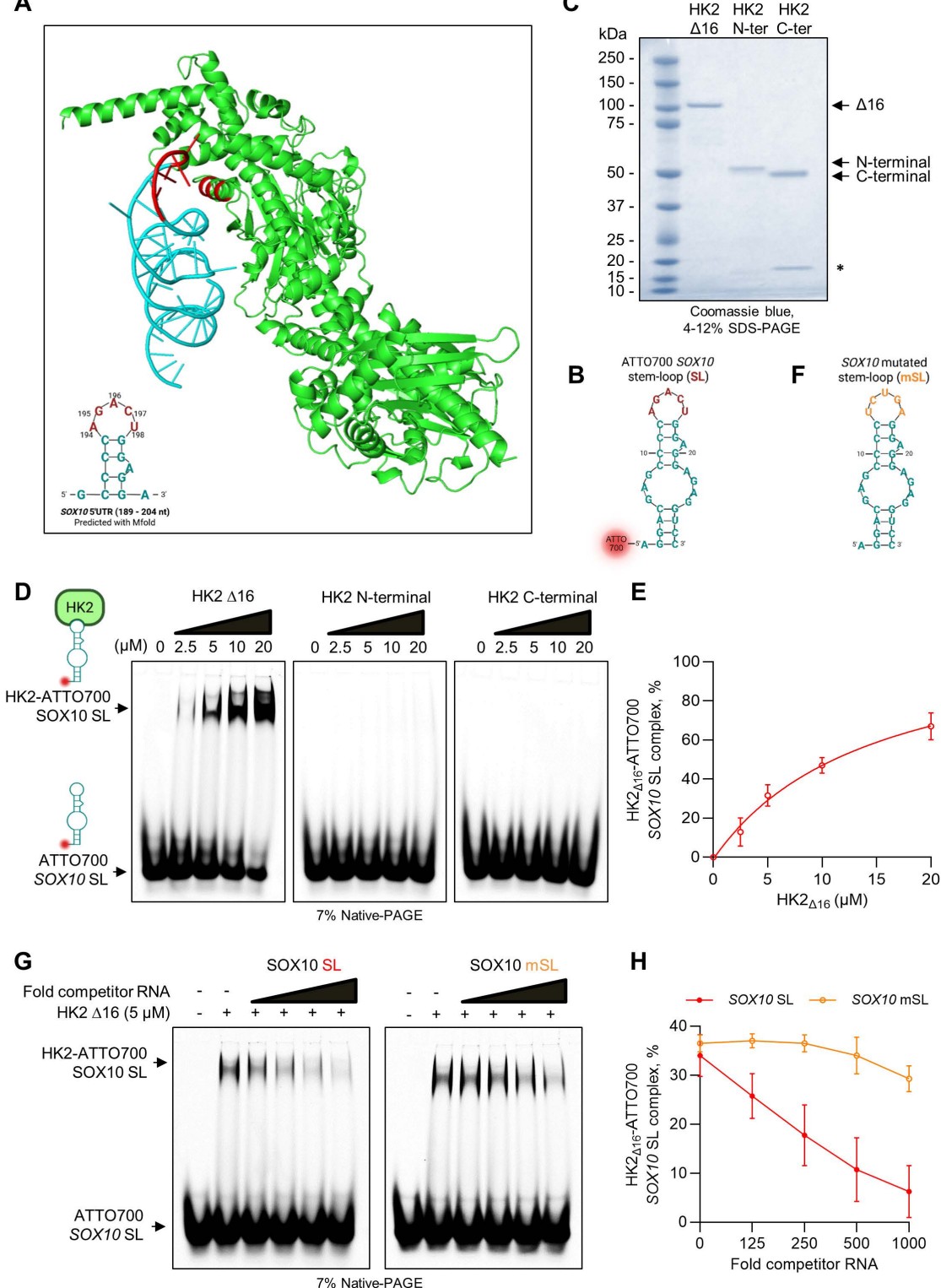

**Fig 5. HK2 binds *SOX10* 5′UTR in vitro. (A)** AlphaFold model of HK2-*SOX10* 5′UTR (189–204 nts) complex. The predicted RNA is shown in cyan, and HK2 protein is shown in green. HK2 residues and *SOX10* mRNA nucleotides at the interaction interface are highlighted in red. Inset: schematic representation of the secondary structure of the *SOX10* 5′UTR (189–204 nts) predicted by Mfold, with the putative HK2-binding region (194–198 nts)

marked in red. **(B)** Secondary structure of the 30-nt 5′-end fluorescently labeled (ATTO700) *SOX10* stem-loop (SL) RNA used in EMSA. **(C)** SDS-PAGE gel showing 0.5 μg of purified recombinant HK2 variants loaded on a 4%–12% Bis–Tris gel, stained with Coomassie blue. A major band is observed in the expected molecular mass of HK2 Δ16 (100.5 kDa), N-terminal (51.3 kDa), and C-terminal (48.5 kDa), as indicated by a black arrow. An asterisk (*) indicates a nonspecific contaminant band. Δ16: HK2 protein deleted of the first 16 amino acids of the N-terminal domain that binds the OMM; N-ter: N-terminal domain of the HK2 protein; C-ter: C-terminal domain of the HK2 protein. **(D)** Representative EMSA showing binding of recombinant HK2 variants (Δ16, C-ter, and N-ter) to ATTO700 *SOX10* SL RNA (25 nM) at the indicated HK2 concentrations. HK2-ATTO700 *SOX10* SL complexes (top) and free ATTO700 *SOX10* SL (bottom) are indicated by black arrows. **(E)** Quantification of EMSA replicates shown in (D). **(F)** Secondary structure of the 30-nt *SOX10* mutated stem-loop (mSL) RNA used as a negative control. **(G)** Competitive EMSA using 5 μM HK2 Δ16 and 25 nM ATTO700-labeled SOX10 SL. Increasing concentrations of unlabeled competitor RNAs (*SOX10* SL or *SOX10* mSL) were added as indicated by black arrows. **(H)** Quantification of EMSA replicates shown in (G). The individual numerical values for panels Fig 5E and 5H are available at S5 Data.

levels (Fig 6A and 6B), glucose starvation resulted in the loss of RIBOmap signals which were recovered after G-6-P supplementation (Fig 6D and 6E), suggesting that *SOX10* mRNA translation is regulated by glucose through HK2 release from the mitochondria. We then investigated whether the interaction between HK2 and the *SOX10* mRNA is regulated in the same manner. This was done using *in situ* RNA-PLA experiments. In line with the RIBOmap results, glucose starvation resulted in a loss of HK2-*SOX10* PLA signals when compared to high-glucose, which was recovered after G-6-P supplementation (Fig 6F and 6G). To confirm that HK2 interacts with the *SOX10* mRNA when HK2 is released from the mitochondria in high-glucose conditions, we co-stained the cells with TOM70, a marker of mitochondria. We found that 78%–88% of HK2–*SOX10* mRNA interactions do not colocalize with the mitochondria (Fig 6H and 6I) and that 71%–82% of translating *SOX10* mRNAs also localize outside of the mitochondria (Fig 6J and 6K).

## Oncogenic functions of HK2 are, at least in part, dependent on the translational regulation of the *SOX10* mRNA

We investigated whether HK2 contributes to cancer-related phenotypes by modulating SOX10 mRNA translation. To this end, we ectopically expressed SOX10 in melanoma cell lines through the transduction of lentivirus particles carrying the sequence of the human SOX10 open reading frame (ORF) fused to Myc-DDK tag, and lacking the 5′ and 3′UTRs (Fig 7A). In the generated cell lines (called ΔUTR SOX10), we observed that HK2 depletion specifically decreases the level of endogenous SOX10, but not the level of Myc-DDK-SOX10 (Fig 7B), thus supporting the requirement of SOX10 5′UTR for the translation regulation mediated by HK2.

We conducted several cell functional assays, such as migration, clonogenic, and proliferation assays.

Using 2D IncuCyte scratch-wound healing assays, we performed migration assays in A375 cells upon stable depletion of HK2 (shHK2). After 72 h of follow-up, we observed that HK2-depleted cells migrated significantly less than their respective parental cells (shCTRL) (S10A Fig). However, ectopic expression of Myc-DDK-SOX10 (ΔUTR) did not reduce the effect observed upon HK2 depletion. Conversely, Myc-DDK-SOX10 (ΔUTR) enhances this effect, thereby indicating that the involvement of HK2 on the migration properties of melanoma cells is independent of the *SOX10* UTR (S10A Fig). Our results are in accordance with different reports demonstrating that SOX10 expression inversely correlates with cell migration [44,45].

Besides, while SOX10 knockdown has been shown to promote cell migration, it has also been demonstrated to decrease melanoma cell proliferation and clonogenicity [28,46,47]. Accordingly, we found that HK2 depletion decreases colony formation of melanoma cells (A375 and A2058) after 10 days of culture (Fig 7C). Ectopic expression of Myc-DDK-SOX10 (ΔUTR), however, reduces the effect of HK2 depletion, suggesting that the effect observed on colony formation of melanoma cells requires, in part, the *SOX10* UTR. Next, we investigated whether HK2 contributes to proliferation properties of melanoma cells via the *SOX10* UTR (Figs 7D and S10B). We found that HK2 depletion decreases cell proliferation of melanoma cells after 6 days. Ectopic expression of Myc-DDK-SOX10 (ΔUTR) in A375 cells rescues the effect of HK2 depletion on proliferation, without having any effect on the proliferation of control cells (siCTRL), indicating that the effect of HK2 depletion on this phenotype requires the *SOX10* UTR. In A2058 cells, ectopic expression of Myc-DDK-SOX10

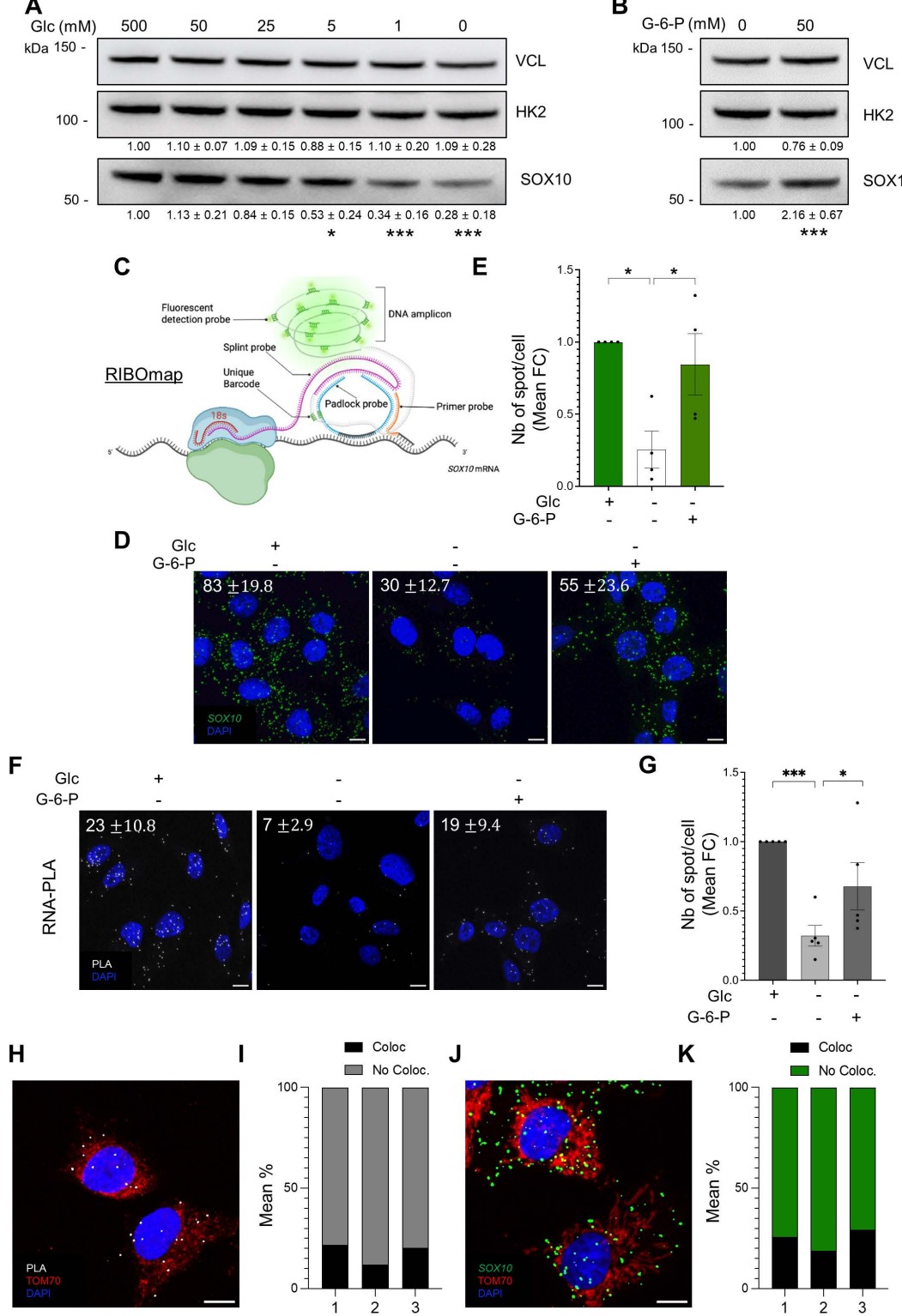

**Fig 6. Glucose and G-6-P levels modulate HK2-*SOX10* mRNA association and *SOX10* mRNA translation.** **(A)** Western blot analysis of HK2 and SOX10 protein level in A375 cells cultured for 24 h with decreasing concentrations of glucose or under glucose starvation, as indicated. HK2 and SOX10 protein quantification is normalized to VCL expression, and the data presented as relative to high glucose (500 mM). *p*-values were calculated

by ordinary two-way ANOVA with Dunnett's multiple comparisons test (SD, *n* = 3 biological replicates) and only significant comparisons are shown (* $p \le 0.05$, *** $p \le 0.001$). **(B)** Western blot analysis of HK2 and SOX10 protein level in A375 cells cultured under glucose starvation for 24 h, with or without subsequent incubation with glucose-6-phosphate (G6P) for 24 h. *p*-values were calculated by ordinary two-way ANOVA with Šídák's multiple comparisons test (SD, *n* = 4 biological replicates) and only significant comparisons are shown (* $p \le 0.05$, *** $p \le 0.001$). **(C)** Schematic of the ribosome-bound mRNA mapping (RIBOmap) used to detect translating *SOX10* mRNAs. RIBOmap relies on the use of a tri-probe set: (1) a primer probe that hybridizes to the target mRNA (i.e., *SOX10* mRNA), (2) a splint DNA probe that hybridizes with the ribosomal 18S RNA, and (3) a padlock probe. When in proximity, the tri-probes produce DNA amplification signals corresponding to active mRNA translation. **(D)** Representative confocal images of translating *SOX10* mRNAs (green spots) in A375 cells cultured with glucose (500 mM) or under glucose starvation for 24 h, with or without subsequent incubation with (G6P) for 2 h. Nuclei are stained with DAPI (blue). The average number of spot/cell ± SD is reported for each condition. Scale bar: 10 μm. **(E)** Quantification of translating *SOX10* mRNAs in the condition described in (D). The data shown represent the mean fold change ± SEM of the number of spots/cells quantified in 3 independent experiments. *p*-values were calculated by paired, two-tailed Student *t* test (* $p \le 0.05$). **(F)** Representative confocal images of *SOX10* mRNA-HK2 protein interaction detected by PLA (white) in A375 cells cultured with glucose (500 mM) or under glucose starvation for 24 h, with or without subsequent incubation with glucose-6-phosphate (G6P) for 2 h. Nuclei are stained with DAPI (blue). The average number of spot/cell ± SD is reported for each condition. Scale bar: 10 μm. **(G)** Quantification of the PLA signal in the condition described in (F) (*n* = 3 biological replicates, mean fold change ± SEM). *p*-values were calculated by paired, two-tailed Student *t* test (* $p \le 0.05$, *** $p \le 0.001$). **(H)** Representative high-magnification confocal images of *SOX10* mRNA-HK2 protein PLA signal *(SOX10* mRNA-HK2 protein interaction, white) and anti-TOM70-stained mitochondria (red) in A375 cells cultured in the presence of glucose (500 mM). Nuclei are stained with DAPI (blue). Scale bar: 10 μm. **(I)** Quantification of the mean % of PLA signal colocalizing or not with mitochondria. **(J)** Representative high-magnification confocal images of RIBOmap signal (translating *SOX10* mRNA, green) and anti-TOM70-stained mitochondria (red) in A375 cells cultured in the presence of glucose (500 mM). Nuclei are stained with DAPI (blue). Scale bar: 10 μm. **(K)** Quantification of the mean % of RIBOmap signal colocalizing or not with mitochondria. The individual numerical values for panels Fig 6E, 6G, 6I, and 6K are available at S6 Data.

(ΔUTR) has an effect in cell proliferation in both control cells (siCTRL) and siHK2-depleted cells, even if the effect is more important in HK2-depleted cells. For this cell line, we cannot rule out that the rescue could be in part due to the transcription activity of SOX10 linked to its overexpression (about 4-fold, Fig 7A).

Collectively, our findings suggest that the oncogenic functions of HK2 in melanoma cells are, at least in part, dependent on the translational regulation of the *SOX10* mRNA via its 5′UTR. Indeed, Myc-DDK-SOX10 (ΔUTR) ectopic expression reduced the effect of HK2 depletion on colony formation, while rescuing the proliferation properties of the melanoma cells. However, HK2 depletion affects melanoma cell migration via other mechanisms, independently of SOX10.

## Discussion

The glycolytic enzyme HK2 has recently been found to perform nonglycolytic activities in cancer, including regulation of transcription [9], anti-apoptotic [48,49], scaffolding activities [14], and protein kinase activities mediating tumor immune evasion [36]. In yeast, hexokinases have been shown to directly bind to nucleic acids [32], thus suggesting unexpected DNA- or RNA-binding activities. Although HK2 does not harbor any recognizable RNA-binding domain, this kinase has been proposed to bind to poly(A) and nonpoly(A) RNAs in human cells in large-scale RNA interactome studies [15–18]. In this work, we demonstrated that HK2 is a novel *bona-fide* RBP implicated in the control of mRNA translation.

To demonstrate that the RNA-binding activity of HK2 sustains its ability to regulate mRNA translation, we focused our study on one given mRNA (*SOX10*) that is also highly relevant in cancer. We found that HK2 specifically binds the *SOX10* mRNA and regulates mRNA translation initiation in a 5′UTR-dependent manner. We found that the same region in the *SOX10* 5′UTR containing a stem-loop structure is essential for both HK2 interaction and mRNA translation. This indicates that HK2 binding to RNA directly sustains its role as a translation regulator.

Given the fact that HK2 regulates mRNA translation of dozens of mRNAs, including the validated *CDH2* and *WNT5A* mRNAs, it is likely that HK2 regulates mRNA translation through its direct binding to these mRNAs. We, however, could not identify in these other mRNAs a sequence/secondary structure motif that is similar to the *SOX10* RNA motif identified in this study. More work, including EMSA and CLIP experiments followed by RNA sequencing, is required to get a more global view of the HK2 RNA binding sites in mRNAs. The structural basis of the specific interaction between HK2 and its RNA targets remains to be defined to better understand how RNA binding affects mRNA translation.

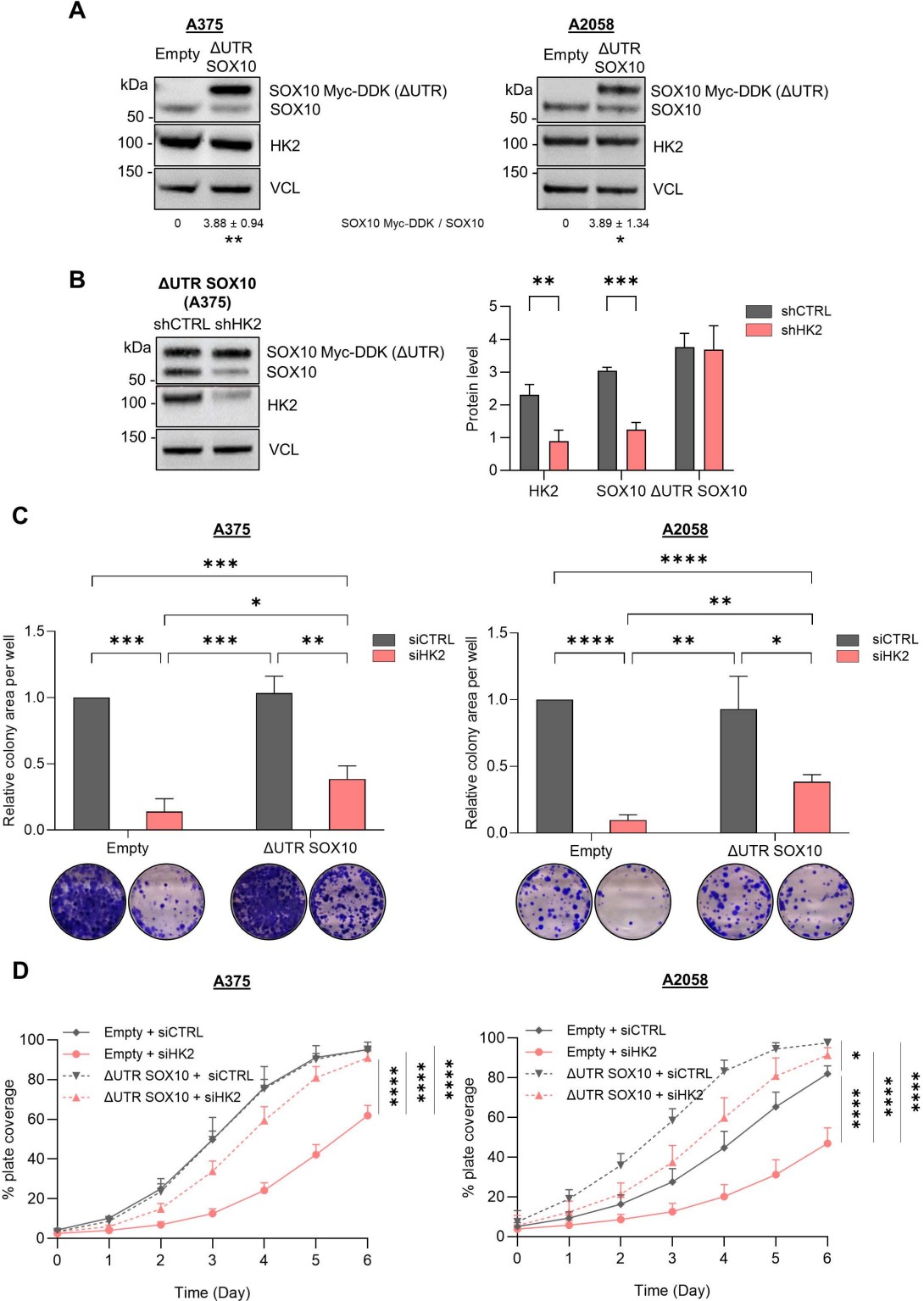

**Fig 7. The effect of HK2 depletion on proliferation and clonogenic properties of melanoma cells requires, at least in part, the *SOX10* UTR.** (A) Representative western blot analysis of the ectopic ΔUTR SOX10 expression in A375 (right panel) and A2058 (left panel) melanoma cells. VCL was used as loading control (*n* = 3 biological replicates). (B) western blot analysis of HK2 and SOX10 protein levels in A375 (right panel) cells upon stable

HK2 knockdown. Quantification of HK2, endogenous SOX10 and ΔUTR SOX10 protein levels are normalized to VCL expression. *p*-values were calculated by ordinary two-way ANOVA with Šídák's multiple comparisons test (SD, $n = 4$ biological replicates) and only significant comparisons are shown (** $p \leq 0.01$, *** $p \leq 0.001$). **(C)** A375 and A2058 cells from (A) were plated 24 h after transfection with siRNAs targeting HK2 (siHK2, light red) or control (siCTRL, gray). The colonies were stained with crystal violet after 10 days, and the clonogenic cell growth was measured. Upper panel: percentage of area covered by crystal violet-stained cell colonies. Lower panel: representative images of three independent experiments. *p*-values were calculated by unpaired, two-tailed Student *t* test (SD, $n = 3$ biological replicates), and only significant differences are shown (* $p \leq 0.05$, ** $p \leq 0.01$, *** $p \leq 0.001$, **** $p \leq 0.0001$). **(D)** Cell proliferation assay performed in A375 and A2058 cells from (A). Cells were plated 24 h after transfection with siRNAs targeting HK2 (Empty, light red; SOX10, dashed light red) or control (Empty, gray; SOX10 dashed gray). *p*-values were calculated by ordinary one-way ANOVA with Turkey's multiple comparison test (SD, $n = 3$ biological replicates), and only significant differences within the same cell line are shown (* $p \leq 0.05$, **** $p \leq 0.0001$). The individual numerical values for panels Fig 7B–7D are available at S7 Data.

Beyond HK2, metabolic enzymes, particularly glycolytic enzymes, have been described to bind to a wide variety of RNA types (e.g., mRNAs, tRNAs, vRNAs), sequences (e.g., AU-rich sequences), regions (5′UTR, 3′UTR, CDS) and secondary structures (e.g., stem loop), holding crucial functions in regulating co- and post-transcriptional steps of gene expression by acting as *trans*-acting factors to control the RNA fate [50]. Some reports have demonstrated the involvement of the RNA-binding activity exerted by GAPDH in the control of mRNA stability through interaction with 3′UTRs of several mRNAs in cancer cells [51–54]. In addition, PKM has been shown to promote ribosome stalling in the vicinity of glutamate- and lysine-encoding regions in ORFs, thereby inhibiting translation elongation [24]. Regarding mRNA translation, the few glycolytic enzymes found involved in this process were described to either interact with ribosomes [21,24] or directly bind to their target RNAs (including tRNAs) [55–59]. Our findings extend these studies, providing new evidence that the moonlighting RNA-binding activity of glycolytic enzymes (*i.e.,* HK2) plays a key role in translational initiation through an interaction with a 5′UTR.

Since (i) regulation of translation initiation is a crucial mechanism for gene expression, dynamically regulating protein synthesis and thereby contributing to the determination of the cellular phenotype [60,61], and (ii) our results demonstrate that the HK2-*SOX10* 5′UTR RNA-protein interaction is involved in cancer-relevant processes such as the ability of melanoma cells to proliferate and form colonies, further research is needed to unravel the factors involved in the regulation of *SOX10* mRNA translation by HK2. This is particularly important since HK2 is an attractive target in cancer [62] and since SOX10 is not only critical in melanoma proliferation but also a factor of resistance to immunologic cell death [63].

While aerobic glycolysis provides significant advantages for rapidly growing and proliferating cells, contributing to 60% of total cellular ATP production in cancer cells [64] and supplying glycolytic intermediates necessary for the biosynthesis of nucleic acids, lipids, and amino acids [4], its role extends further. Given that many glycolytic enzymes also serve as RBPs engaged in post-transcriptional RNA-processing reactions and considering that aberrant mRNA translation is a hallmark of tumors, it is crucial to investigate the significance of HK2 interactions with RNA beyond the *SOX10* mRNA. HK2 might regulate the translation of several other key mRNAs involved in cancer-relevant signaling pathways. Understanding the cellular processes influenced by HK2 RNA-binding activity and uncovering their underlying molecular mechanisms represent promising directions for future research. Such studies could lead to the development of novel therapeutics that specifically block HK2-RNA interaction in cancer cells while sparing nonmalignant counterparts, offering new perspectives in cancer treatment.

Finally, while we found that HK2 interacts with the *SOX10* mRNA to regulate its translation, it is also possible that RNA interaction with HK2 modulates its protein activity. In this context, RNA has been described to mediate the inhibition of the enzymatic activity of the glycolytic enzyme ENO1 in the context of stem cell differentiation [65]. Future work will have to carefully examine the possible RNA-mediated modulation of the enzymatic activity (riboregulation) of the glucose kinase HK2.

## Materials and methods

### Cell culture, transfection, and transduction

M249 cells were grown in RPMI 1,640 (Biowest, #L0500) supplemented with 10% FBS (Gibco, #A5256701) and 2 mM ʟ-glutamine (Eurobio-Scientific, #CSTGLU00-0U). MALME 3M cells were grown in high glucose (4.5 g/L) DMEM

supplemented with 20% FBS. HEK293T, A375, A258 and other melanoma cells were grown in high-glucose DMEM (Biowest, #L0101-500) supplemented with 10% FBS and 2 mM L-glutamine. All cells were grown in 5% $CO_2$ humidified incubator at 37°C. When indicated, the high glucose medium of A375 cells was replaced with glucose-free DMEM (Gibco, #11,966,025) containing 10% FBS and 2 mM L-glutamine and supplemented with 5 g/L galactose (Sigma-Aldrich, #G5388) or 16 mM 2-DG (MedChemExpress, #HY-13966) when indicated. The cells were washed with PBS before the medium was replaced. Cells were regularly tested for mycoplasma and authenticated by short tandem repeat (STR) profiling.

Cells were plated one day before for siRNA and plasmid transfections. The siRNAs were purchased from Dharmacon (ON-TARGETplus technology), and siRNA transfections were performed using Lipofectamine RNAiMAX (Invitrogen, #13,778,150) according to manufacturer's instructions. Transfected cells were either harvested at 48 h or re-plated 24 h after transfection, as indicated in the figure legends. Plasmid transfections were performed using Lipofectamine 2000 (Invitrogen, #11,668,019) or JetPEI (Polyplus Transfection, #101,000,020), and the transfection was performed according to manufacturer's instructions. At 48 h post-transfection, cells were harvested for further analysis. Lipofectamine 2000 was used for the transfection of Renilla and Firefly luciferase reporters in A375 cells. The cDNA for human *SOX10* 5′UTR and 3′UTR were subcloned into the pRP(Exp)-EGFP-CMV vector upstream and downstream a RLuc sequence, respectively, and they were kindly provided by Prof Caroline Robert (Gustave Roussy, France). Site-directed mutagenesis of *SOX10* 5′UTR was performed using Q5 Site-Directed Mutagenesis Kit (New England Biolabs, #E0554) according to the manufacturer's instructions. JetPEI was used for transient transfection of HK2 plasmids in HEK293T cells. HK2 (NM_000189.5) plasmids were purchased from VectorBuilder. HK2-WT (wild-type; #VB220927−1092xyz), DA (D209A/D657A, a nonglucose-binding mutant; #VB221214−1257apk), SA (S340A/S788A, a catalytically inactive mutant; #VB221214−1258ubs), and mitochondrial-binding deficient (MBD) (deletion of 1–15aa, MBD; #VB221214−1255ezg) were cloned into a pLV[Exp]-Puro-CMV EGFP expressing vector.

To establish the stable expression of ΔUTR SOX10 (Myc-DDK tagged; OriGene, #RC203545L3V) in A375 and A2058 cell lines, cells were transduced with pLenti-C-Myc-DDK-P2A-Puro vector expressing SOX10 (NM_006941) ORF nucleotide sequence (OriGene, #RC203545L3V) according to manufacturer's instructions. A375 cells with stable HK2 knockdown were generated after virus production and infection using a lentiviral vector (VB220929−1200hug) expressing a human HK2 hairpin target sequence (5′-CTTAGGGCAGTCAGTAGTATTCTCGAGAA TACTACTGACTGCCCTAAG-3′), also according to manufacturer's instructions. Lentivirus infection was performed during 48 h, and SOX10 overexpressing and HK2 knockdown cells were selected with puromycin (2 µg/ml) and hygromycin B (400 µg/mL), respectively. Western blot and RT-qPCR analyses were performed to confirm the ectopic expression of ΔUTR SOX10 and HK2 knockdown in the stable cells. The established cell lines were cultured in high glucose DMEM medium with 10% FBS and 2 mM L-glutamine supplemented with puromycin (2 µg/ml) or hygromycin B (400 µg/mL).

**Polysome profiling**

Polysome profiling was performed as previously described [66,67]. Briefly, A375 cells were incubated with 100 µg/mL cycloheximide in fresh medium at 37°C for 5 min. Cells were then washed, scraped into ice-cold PBS supplemented with 100 µg/mL cycloheximide (Sigma-Aldrich, #C4859), and centrifuged for 10 min at 3,000 rpm. The cell pellets were resuspended into 400 µL of LSB buffer (20 mM Tris, pH 7.4, 100 mM NaCl, 3 mM MgCl2, 0.5 M sucrose, 1 mM DTT, 100 U/mL RNasOUT and 100 µg/mL cylcoheximide). After homogenization, 400 µL LSB buffer supplemented with 0.2% Triton X-100 and 0.25 M sucrose was added. Samples were incubated on ice for 30 min, and then centrifuged at 12,000$g$ for 15 min at 4°C. The supernatant was adjusted to 5 M NaCl (Sigma-Aldrich, #S6546) and 1 M MgCl2 (Invitrogen, AM9530G). The lysates were then loaded onto a 5%–50% sucrose density gradient and centrifuged in an SW41 Ti rotor (Beckman) at 36,000 rpm for 2 h at 4°C. Polysome fractions were monitored and collected using a gradient fractionation system (Isco). The sucrose gradient fractions were stored at −80°C or directly processed for western blot analysis or RNA extraction. For RT-qPCR analysis, RNAs from each fraction were extracted using TRIzol-LS (Invitrogen, #10,296,028) according to

manufacturer's procedure and were quantified using RNA 2,100 Bioanalyzer (Agilent Genomics). The expression of a panel of 84 EMT-associates genes was measured using RT$^2$ Profiler PCR Array (qPCR Qiagen EMT array, #330,231) in both total RNA and polysomal RNA levels of A375 cells transfected with siRNAs targeting HK2 or control.

## RNA sequencing and bioinformatic analysis

RNA sequencing libraries were prepared from 500 ng to 1 μg of total RNA or mRNA-enriched from heavy polysome fractions using the Illumina TruSeq Stranded mRNA Library preparation kit which allows to perform a strand-specific sequencing. Nano-drop spectrophotometer was used to assess purity of RNA based on absorbance ratios (260/280 and 260/230) and BioAnalyzer for RNA integrity (RIN > 9). A first step of polyA+ selection using magnetic beads is done to focus sequencing on polyadenylated transcripts. After fragmentation, cDNA synthesis was performed and resulting fragments were used for dA-tailing followed by ligation of TruSeq indexed adapters. PCR amplification was finally achieved to generate the final barcoded cDNA libraries. The libraries were equimolarly pooled and subjected to qPCR quantification using the KAPA library quantification kit (Roche). Sequencing was carried out on the NovaSeq 6,000 instrument from Illumina based on a 2*100 cycle mode (paired-end reads, 100 bases) to obtain around 30 million clusters (60 million raw paired-end reads) per sample. Finally, Fastq files were generated from raw sequencing data using bcl2fastq pipeline performing data demultiplexing based on index sequences.

RNA-seq data were analyzed with the Institut Curie RNA-seq Nextflow pipeline [68] (v3.1.4). Briefly, raw reads were first trimmed with Trimgalor. Reads were aligned on the human reference genome (hg19) using STAR (STAR_2.6.1a_08–27). Genes abundances were then estimated using STAR, and the Gencode v34 annotation. Differential analysis between conditions were done using the R package Xtail [69] only on protein-coding genes.

We used the variance stabilizing transformation (VST) offered by *DESeq2* [70] on the raw count data to stabilize the variance across the mean and then performed a principal components analysis (PCA). The PCA plots (S11 Fig) were built using the ggplot2 package [71].

## Complex Capture (2C)

A375 cells were irradiated with UV-C light at 254 nm and lysed in HMGN150 buffer (20 mM HEPES pH 7,5; 150 mM NaCl; 2 mM MgCl$_2$; 0.5% Nonidet P-40; 10% Glycerol). Lysates were cleared via centrifugation at 10,000*g* for 10 min at 4°C, and then treated or not with RNAse A (15 μL of RNAse A at 10 mg/mL for 1 mg of proteins; Thermo Scientific, #EN0531). A fraction of the input (2%) was kept as control for the protein expression on SDS-PAGE. The Quick-RNA Midiprep kit was to purify crosslinked nucleic acids–RBP complexes. RNA concentration of the eluate was measured using NanoDrop (Thermo Scientific), and 50 μg RNA was treated with RNAse I (500 U; Invitrogen, #AM2295) for 40 min at 30°C. The samples were then mixed with 1× loading buffer for subsequent capillary western blot analysis.

## CrossLinking and immunoprecipitation (CLIP)

Cells were cultured on a 15 cm dish until 70%–80% of confluence. HK2 siRNA-mediated depletion was performed where indicated. The medium was removed, and the cells were washed with ice-cold PBS and irradiated with UV-C light at 254 nm. The cells were then lysed in 1 mL RIPA lysis buffer (50 mM Tris-HCl pH 7.4, 100 mM NaCl, 1% Nonidet P-40, 0.1% SDS, 0.5% Sodium deoxycholate), supplemented with RNAseOUT (40 U/mL; Invitrogen, #10,777,019) and protease inhibitor cocktail (complete EDTA free; Roche, #11,873,580,001). Lysates were cleared via centrifugation at 18,000 g for 10 min at 4°C, and then treated or not with RNAse I (0.3–1 U) in combination with TURBO DNase I (4 U; Invitrogen, #AM2238) and incubated at 37°C for 3 min at 800 rpm. 1% of the lysate was used as input material to quantify total protein concentration in western blot analysis. The rest of the lysate was incubated with the HK2 antibody-conjugated beads or GFP-Trap Magnetic Agarose (Chro-moTek, #gtma) overnight at 4°C while constantly rotating. For endogenous HK2 immunoprecipitation, 9 μg of HK2 antibody (Abcam, ab209847) or appropriate control IgG (rabbit; Cell Signaling, #2729S) were coupled overnight at 4°C to 90 μL

Dynabeads Protein G (Invitrogen) while constantly rotating, and prior to cell harvesting. Conversely, GFP-Trap Magnetic Agarose was used for the immunoprecipitation of HK2 mutants conjugated with GFP. The immunocomplexes were washed twice with RIPA-S buffer (50 mM Tris-HCl pH 7.4, 1 M NaCl, 1 mM EDTA, 1% Nonidet P-40, 0.1% SDS, 0.5% Sodium deoxycholate) and twice with PNK buffer (20 mM Tris-HCl pH 7.4, 10 mM MgCl$_2$, 0.2% Tween 20). After washes, 10% of the immunocomplexes was used to validate the correct immunoprecipitation by western blot analysis. Then, crosslinked nucleic acids were radiolabeled with ATP, [γ-32P] (PerkinElmer) by T4 PNK (New England Biolabs, #M0201S) for 30 min at 37°C. After two washes with RIPA-S buffer and one wash with PNK buffer, protein-RNA complexes were eluted in 1× loading buffer at 75°C for 10 min and separated by SDS-PAGE. For RNA extraction experiments (i.e., S5B Fig), we used liquid transfer to transfer the RNA-protein complexes from the gel to a nitrocellulose membrane at 100 V for 2 h (1× transfer buffer, 20% ethanol). The nitrocellulose membrane was cut at the expected size of the HK2-RNA complexes, follow by 10 μL proteinase K (20 mg/mL; Invitrogen, #25,530,049) digestion first in 200 μL PK buffer (100 mM Tris-HCl pH 7.4, 50 mM NaCl, 10 mM EDTA) and then in 200 μL PK-urea buffer (100 mM Tris-HCl pH 7.4, 50 mM NaCl, 10 mM EDTA, 7M Urea), both at 1,000 rpm for 20 min at 45°C. The RNA was phase-separated using ~410 μL phenol:chloroform:isoamyl alcohol (125:24:1; Sigma, P1944) (1:1), after 5 min centrifugation at 15,000$g$ at room temperature. The RNAs were then precipitated at −20°C overnight by adding a mix of 40 μL of sodium acetate pH 5.5 (3M; Invitrogen, #AM9740), 1 μL of glycoblue, and 1 mL of absolute ethanol. The next day, samples were centrifuged, supernatant removed, and pellets washed with 1 mL of 80% ethanol. Pellets were resuspended in RNAse-free water and incubated for 5 min in 1× RNA loading dye (New England Biolabs, #B0363S) at 80°C at 1,000 rpm. Samples were loaded onto a denaturing 6% TBE-urea gel and migrated at 70 V for 1 h 30 min in 1× TBE migration buffer. Then the gel was dried, and RNAs or, for standard CLIP, RNA-protein complexes were exposed to a phosphorimaging screen and scanned with Amersham Typhoon Biomolecular Imager (CYTIVA). The files were processed with ImageJ software.

## Sucrose density gradient ultracentrifugation and fractionation

For the sucrose density gradient ultracentrifugation and fractionation assay, previously published protocols were used as a basis [38,65]. A375 cells were cultured on a 15 cm dish and lysed in 300 μL lysis buffer (25 mM Tris-HCl pH 7.4, 2 mM EDTA, 150 mM KCl, 1 mM NaF, 0.5 mM DTT, 0.5% Nonidet P-40), supplemented with protease inhibitor cocktail (complete EDTA-free; Roche, #11,873,580,001). As control, lysis buffer was also supplemented with RNAseOUT (40 U/mL; Invitrogen, #10,777,019). Lysates were pre-cleared via centrifugation at 10,000$g$ for 10 min at 4°C. The lysates were then treated with a combination of RNase I (150 U; Invitrogen, #AM2295) and RNase A/T1 (50 U/ 125 U; Thermo Scientific, #EN0551) and incubated at 37°C for 15 min. For the control sample, RNaseOUT was added to the lysate and the sample was incubated for 15 min at 4°C. For the fractionation, sucrose gradients were prepared from 5% (w/v) to 25% (w/v) sucrose (in 150 mM KCl, 25 mM Tris-HCl pH 7.4 and 2 mM EDTA) using Isco Model 160 Gradient Former. Lysates were then separated by centrifugation at 30,000 rpm and 4°C in an SW41Ti rotor (Beckman) for 18 h. After ultracentrifugation, the lysate fractions were monitored and carefully collected using a gradient fractionation system (Isco) (20 fractions were collected of approximately 1,000 μL) and transferred into fresh 1.5 mL tubes. For the protein precipitation, 150 μL of cold 100% Trichloroacetic acid (TCA; Sigma-Aldrich, #T6399) was added, and the samples were incubated on ice for 30 min. The fractions were centrifuged at 18,000$g$ for 20 min at 4°C. The TCA supernatant was carefully removed, and the pellets were washed once with 1 ml cold acetone (stored at −20°C). The fractions were vortexed, and an additional centrifugation step was performed at 18,000$g$ for 30 min at 4°C. The supernatant was carefully removed, and the pellet air-dried. Finally, the pellets were resuspended in 20 μL 1× loading buffer, incubated at 95°C for 5 min, and used for SDS-PAGE and immunoblotting.

## RNA immunoprecipitation (RIP) assay

Prior to harvesting cells, 9 μg of antibody anti-HK2 (Abcam, ab209847) were coupled overnight at 4°C to 90 μL Dynabeads Protein G (Invitrogen) while constantly rotating. A375 and A2058 cells were cultured on a 15 cm dish and lysed in 500 μL Co-IP buffer (50 mM Tris-HCl pH 7.4, 1 mM EDTA, 150 mM NaCl, 0.1% Nonidet P-40), supplemented with protease inhibitor

cocktail (complete EDTA free; Roche, #11,873,580,001) and RNAseOUT (40 U/mL; Invitrogen, #10,777,019) for 30 min at 4°C. Lysates were cleared via centrifugation at 18,000g for 10 min at 4°C. After lysis, 1% and 10% of the lysate was used as input material to confirm total protein and RNA levels, respectively. The rest of the lysate was incubated with the HK2 antibody-conjugated beads overnight at 4°C while constantly rotating. The immunocomplexes were washed five times with ice-cold CO-IP buffer supplemented with protease inhibitor cocktail and RNAseOUT. After washes, 10% of the immunocomplexes was used to validate the correct immunoprecipitation by western blot analysis. The rest of the beads were eluted with 100 μL Protein-RNA elution buffer (100 mM Tris-HCl pH 8, 10 mM EDTA, 1% SDS) at 80°C for 5 min, followed by another 5 min incubation at room temperature. To recover the RNA bound by HK2, the beads were resuspended in TRIzol (Invitrogen, #15,596,018), as well as the input RNA, and RNA extraction was followed according to manufacturer's procedure. The extracted RNA was then used for subsequent RT-qPCR experiments. The efficiency of the primers was validated, and the Ct value of each RIP RNA fractions had been adjusted to the efficiency of the primers to ensure consistency.

### In silico binding prediction

The catRAPID algorithm [39] was used to estimate the potential binding sites of HK2 on *SOX10* mRNA. The highest-ranking site at RNA position 206–321 bp was used for subsequent analysis.

### Recombinant protein expression and purification

The recombinant HK2 proteins were expressed and purified as described previously [37,72]. Δ16-HK2 represents the HK2 protein deleted of the first 16 amino acids of the N-terminal domain (Δ16) that binds the outer mitochondrial membrane (OMM). The individual N-terminal (residues 16–479) and C-terminal domains (residues: 480–917) have been expressed and purified along with the Δ16 of HK2. Briefly, the HK2 variants were cloned into the pET28b bacterial expression vector and expressed in *Escherichia coli* BL21-CodonPlus-RIL (Stratagene). The cell lysate was loaded on Fast Flow Ni-NTA resin followed by a HiLoad Superdex 200 column on an ÄKTA purifier core system (Cytiva Life Sciences, Marlborough, MA, USA). The column was pre-equilibrated with filtration buffer (50 mM Hepes, pH 7.4, 150 mM NaCl, and 0.5 mM tris (2-carboxyethyl)phosphine [TCEP]). The final protein was concentrated to ~50 μM, as determined by the Bradford assay and assessed for purity using SDS-PAGE (Fig 5B).

### Electromobility shift assay (EMSA)

For all EMSA experiments, appropriate concentrations of each recombinant HK2 protein were first incubated with 0.125 mg/mL Yeast tRNA (10 mg/mL; Invitrogen, #AM7119) in binding buffer (10 mM Tris-HCl, 1 mM EDTA, 100 mM KCl, 0.1 mM DTT, 5% glycerol, 0.01 mg/mL BSA, pH 7.5) for 20 min at 20°C. In competition experiments, the reaction mixtures also contained excess of unlabeled competitor RNAs (Eugentec) (S3 Table). Next, the reaction mixtures were incubated with 25 nM of ATTO700-labeled oligonucleotides (IDT) (S3 Table) for 5 min at 20°C in a final volume of 20 μL. The samples were then loaded onto freshly prepared 7% polyacrylamide (37.5:1 acrylamide/bis-acrylamide, 1× TBE, 0.1% APS, 0.1% TEMED) native-PAGE gels, which were run at 45 V for 2 h 30 min at 4°C. Gels were imaged using an Odyssay scanner, and the images were processed with ImageJ software.

### Seahorse metabolic flux assay

OCRs and extracellular acidification rates (ECAR) were analyzed on a XF96 Extracellular Flux Analyzer (Seahorse Bioscience). A375 cells were plated in nonbuffered DMEM medium supplemented with 25 mM glucose (Sigma-Aldrich, #D9434) or galactose (Sigma-Aldrich, #G5388). Measurements were obtained under basal conditions (no treatment) and after the addition of mitochondrial inhibitors (oligomycin, 1 μM; FCCP, 0.5 μM; rotenone/antimycin A, 0.5 μM), or glycolysis inhibitor (2-DG, 16 mM) (MedChemExpress, #HY-13966).

## Lactate measurement assay

Lactate concentration in A375 cells was measured using the Lactate-Glo kit (Promega, #J5021) according to the manufacturer's instructions. Cells were seeded in duplicates in six-well plates and cultured in media supplemented with 10% dialyzed FBS (Gibco, #26,400−036) and 2 mM L-glutamine. Wells with medium only were used as controls. Next day, cells were transfected with 25 nM siRNA (Dharmacon) targeting HK2 or control. After 48 h of transfection, 5 µL of media was removed and diluted in 95 µL PBS. Fifty µL of the diluted samples were mixed with 50 µL of the lactate detection reagent (1:1) for 30–60 s. The mix was incubated for 1 h at RT before recording the luminescence using Berthold TriStar$^2$ LB 942 Microplate Reader.

## Colony formation assay

Cells transfected with 25 nM siRNA (Dharmacon) were seeded in triplicates in six-well plates. After 10 days of incubation, cells were washed with PBS 1× (Gibco, #70,011,044) and stained with 0.5% of crystal violet (Sigma Aldrich, #C0775) for 15 min. Plates were washed with water and dried before imaging. Colony area was quantified using ImageJ-plugin "ColonyArea" [73].

## Luciferase assay

A375 and A2058 cells were transiently transfected with Firefly and Renilla luciferase reporters using Lipofectamine 2000 Reagent (Invitrogen, #11,668,019). Renilla vectors containing the full-length (FL) *SOX10* 3′UTR or 5′UTR, as well as six *SOX10* 5′UTR mutants (Δ1–6) harboring deletions of ~50 nucleotides, were transfected in the cells. An empty Renilla vector (without *SOX10* 5′ or 3′UTR) was used as control for the experiment. After 48 h, part of the transfected cells was harvested for RNA extraction to measure the vectors' RNA expression by qPCR, while the rest of the cells were used to measure the luciferase activity using the Dual-Luciferase Reporter Assay System (Promega). Data was presented as the ratio between Renilla and Firefly luciferase RNA expression or luminescence activities, respectively.

## Proliferation assay

Cells transfected with 25 nM siRNA (Dharmacon) were seeded in triplicates in six-well plates. The cell growth was analyzed for percent plate coverage with IncuCyte S3 Live Cell Analysis System (Sartorius). Pictures of each well were acquired every 3 h for 6 days.

## Scratch-wound assay

Using 2D IncuCyte scratch-wound healing assays, we conducted migration assays in A375 cells upon HK2 knockdown and in cells that ectopically expressed SOX10 (ΔUTR SOX10). Cells were grown to confluence in a pre-coated 96-well ImageLock plate (Sartorius, #BA-04856) in a standard $CO_2$ incubator. Once confluent, the cells monolayers were scratched using the 96-pin IncuCyte WoundMaker tool (Sartorius, #4,563), which simultaneously creates wounds in all the wells. After wounding, wells were washed twice with PBS and appropriate media was added. The cells and gap area were photographed every 3 h, and the wound healing was analyzed and measured by Relative Wound Density (%) using the IncuCyte Live Cell Analysis System. Significance was calculated by comparing HK2 knockdown to parental cells after 72 h.

## Protein extracts, SDS-PAGE, and western blot

Whole cell lysates were prepared using RIPA buffer supplemented with RNaseOUT (40 U/mL; Invitrogen, #10,777,019) and protease inhibitor cocktail (complete EDTA free; Roche, #11,873,580,001). Lysates were cleared via centrifugation at 18,000 *g* for 10 min at 4°C. Cell lysates were quantified for protein concentration using a bicinchoninic acid (BCA) protein

assay kit (Thermo Scientific, #23,225). Protein samples were resolved on NuPAGE 4%–12% Bis–Tris gels with MOPS buffer or 3%–8% Tris-acetate gels with Tris-acetate buffer (Life Technologies, #LA0041) and then transferred to 0.45 µm nitrocellulose membrane (Amersham). After saturation in Tris-buffered saline buffer containing 0.01% Tween-20 (TBST) and supplemented with 5% powdered milk, the membranes were incubated with primary antibodies either for 1 h at RT or overnight at 4°C with agitations, followed by 3 × TBST washes of 15 min each, and incubation of the membranes with peroxidase (HRP)-conjugated secondary antibodies diluted in 5% milk TBST for 1 h at RT. After 3 × TBST washes of 5 min each, the membranes were incubated in ECL solution for development.

The following antibodies were used for assays: anti-HK2 (Abcam, #ab209847; 1:1,000 dilution), anti-SOX10 (Santa Cruz Biotechnology, #sc365692; 1:1,000 dilution), anti-GFP (Roche, #11,814,460,001; 1:1,000 dilution), anti-HUR (Santa Cruz Biotechnology, #sc-5261; 1:1,000 dilution), anti-H3 (Abcam, #ab1791; 1:500 dilution), anti-GAPDH (Sigma-Aldrich, #G9545; 1:5,000), anti-PKM2 (Cell Signaling, #4053S; 1:1,000 dilution), anti-LDHB (Santa Cruz Biotechnology, #sc100775; 1:1,000 dilution), anti-S6 ribosomal protein (Cell Signaling, #2,317; 1:1,000 dilution), anti-VCL (Abcam, #ab129002; 1:3,000 dilution), anti-alpha Tubulin antibody (GeneTex, #GTX628802; 1:1,000 dilution), anti-HSC70 antibody (Santa Cruz Biotechnology, #sc-7298; 1:1,000 dilution), anti-ACOT1/2 (Santa Cruz Biotechnology, #sc-373917; 1:1,000 dilution), goat anti-rabbit (Sigma-Aldrich, Thermo Fisher Scientific, 1:2,000 dilution) and goat anti-mouse (Sigma-Aldrich, Thermo Fisher Scientific, 1:5,000 dilution) antibodies.

### RNA extraction and RT-qPCR

RNA was isolated using the TRIzol-chloroform method and treated with DNase I (TURBO DNA-free; Invitrogen, #AM1907) according to the manufacturer's instructions. After DNase treatment, RNA was quantified using a Nanodrop 2000 spectrophotometer (Thermo Scientific), and 1 µg total RNA was used to synthesize cDNA using SuperScript III Reverse Transcriptase (200 U/µL; Invitrogen, #18,080,044) and random hexamer primers (Invitrogen, #SO142), according to the manufacturer's instructions. Quantitative PCR (qPCR) was performed using the Power SYBR Green PCR Master Mix (Thermo Scientific, #4,367,559) on a CFX96 Real-Time PCR Detection System (Bio-Rad Laboratories). Primer sequences are given in S3 Table. Gene expression values were determined relative to VCL and are shown as a relative fold change to the value of control samples. All experiments were performed in biological triplicates and error bars indicate ± standard deviation as assayed by the ΔΔCt method.

For RT-qPCR analysis of polysome fractions, RNA of each fraction (250 µL), comprising both monosome fractions and polysome fractions, was extracted using TRIzol LS (Invitrogen, #10,296,028). The extracted RNA was diluted in 30 µL RNase-free water (Invitrogen, #10,977,035). For each fraction, the same volume of RNA was retrotranscribed into cDNA using SuperScript IV Reverse Transcriptase (Invitrogen, #18,090,010) and random hexamer primers (Invitrogen, #SO142), according to the manufacturer's instructions. Following the reverse transcription, mRNA abundance was determined by qPCR using Power SYBR Green PCR Master Mix (Thermo Scientific, #4,367,559). Data were analyzed by the threshold cycle (Ct) comparative method and quantified as percentage of the total RNA considering the whole fractions stand for 100%. HPRT gene was used as a control.

### RNA Proximity Ligation Assay (RNA-PLA)

Interactions between HK2 and SOX10 mRNA were monitored *in cellulo* by RNA-PLA [40]. The experiment was performed using Duolink In situ Detection Reagents FarRed according to the manufacturer's instructions (Sigma-Aldrich, #DUO92013). A375 cells were seeded on coverslips and, when applicable, transfected with siRNAs targeting HK2 (siHK2), SOX10 (siSOX10), or control (siCTRL). After 48 h, cells were fixed with 1.6% PFA in PBS for 20 min at RT. To test the HK2 catalytic substrate and product-dependent effect on its proximity to the *SOX10* mRNA, cells were grown in high-glucose-containing media (500 mM) or under glucose starvation for 24 h. Glucose-6-phosphate (50 mM) was added to the media without glucose for 2 h. At the end of the incubation, cells were fixed with 1.6% PFA in PBS for 20 min at RT.

Next, coverslips were washed with PBS and permeabilized with Tween 0.2% in PBS for 10 min at RT. Cells were then washed with PBS and blocked using a blocking buffer (10% goat serum and 0.1% Triton X-100 in PBS supplemented with 20 µg/mL of salmon sperm) for 20 min at 37°C. Cells were once again washed with PBS and incubated in blocking buffer containing 100 nM of an anti-sense PLUS probe in a wet chamber at 4°C overnight. The solution was boiled for 3 min at 70°C prior to addition. Of note, this probe was composed of a 20-mer of DNA complementary to the SOX10 5′UTR, a polyA linker and a sequence complementary to the PLA MINUS probe:

GCGGTCCAGCTCGGGGCTGGGAGGTGACGCTGGTGGGCTGGGAGGGAAAAAAAAAAAAAAAAAAAAATATGA-CAGAACTAGACACTCTT.

Subsequently, cells were washed twice with PBS and incubated in blocking buffer for 1 h at RT. Then, HK2 primary antibody (Abcam, #ab209847) was diluted in blocking buffer (1:1,000) and incubated for 1 h at 37°C. Next, the Duolink PLA probes anti-Rabbit MINUS (5×) containing secondary antibodies conjugated to oligonucleotides were added, and the mixture was incubated for 1 h at 37°C. The ligation reaction was carried out for 45 min at 37°C using Duolink ligase in the supplied ligation buffer. Amplification was then performed for 2 h at 37°C using Duolink polymerase and amplification buffer containing FarRed fluorophore. When applicable, cells were incubated with anti-TOM70 primary antibody (1:400; Proteintech, #66593–1-Ig) for 1 h at 4°C. After washing, cells were incubated for 1 h at RT with an Alexa Fluor 488–conjugated secondary antibody (1:1000; Invitrogen, #A28179). Both antibodies were diluted in PBS containing 5% BSA. Nuclei were stained with DAPI (1:3000) for 30 min at RT. Finally, coverslips were mounted using ProLong Diamond Antifade Mountant (Invitrogen, #P36965). Images were acquired using Leica SP8 confocal microscope, using the appropriate filters at the PICT-IBiSA Imaging Facility in Orsay. At least 30 cells per sample were imaged, and the images were processed with ImageJ by using the MIC-MAQ plugin.

**RIBOmap assay**

RiboMap assay was performed as previously described [43]. The Splints probes, hybridizing with the 18S RNA, were designed by the previous study [43]. The SOX10-specific hybridization regions in Padlocks and Primers probes were designed using Picky 2.2 software. To visualize translating mRNAs, a fluorescent probe complementary to the DNA amplicons was used. Probes were purchased from IDT and probes sequences are listed in (S3 Table).

Briefly, cells seeded on coverslips were transfected with control siRNA or siRNA targeting *SOX10* as described above. Alternatively, cells were glucose-starved or incubated with G-6-P as mentioned above. Cells were fixed with 1.6% PFA in PBS for 20 min at RT. cells were permeabilized with Methanol for 1 h at −20°C. Cells were then washed with PBS and incubated for 10 min with Quenching solution (1 mg/mL yeast tRNA, 0.1 U/µL SUPERase·In RNase Inhibitor, 100 mM Glycine, 0.1% Tween-20 in PBS). After washes, cells were incubated overnight at 40°C in a humidified oven with Hybridization buffer (2× SSC, 10% formamide, 20 mM Ribonucleoside vanadyl complex, 0.1 mg/mL yeast tRNA, 1 nM of pooled padlock and primer probes, 100 nM splint probes, 0.1% Tween-20 in PBS). The day after, cells were washed twice at 37°C for 10 min with PBSTR (0.1 U/µL SUPERase·In RNase Inhibitor, 0.1% Tween-20 in PBS), and once for 20 min with 4× SCC dissolved in PBSTR solution. Cells were then washed for 10 min at RT with PBSTR and incubated for 2 h at RT with the Ligation mixture (0.25 U/µL T4 DNA ligase, 0.5 mg/ml BSA, 0.4 U/µL SUPERase·In, RNase Inhibitor, 1× T4 DNA ligase buffer in water). After washes, with PBSTR, cells were incubated first for 30 min at 4°C and then for 2 h at 30°C with Rolling circle amplification mixture (0.5 U/µL Phi29 DNA polymerase, 250 µM dNTPs, 0.5 mg/ml BSA, 0.4 U/µL SUPERase·In RNase Inhibitor, 1× Phi29 buffer in water). Cells were then washed with PBS, incubated for 30 min with 50 nM of fluorescent detection probe and DAPI (1:3,000) and mounted using ProLong Diamond Antifade Mountant (Invitrogen, #P36965). When applicable, before mounting cells were incubated with anti-TOM70 primary antibody (1:400; Proteintech, #66593–1-Ig) for 1 h at 4°C. After washing, cells were incubated for 1 h at RT with an Alexa Fluor 594–conjugated secondary antibody (1:1000; Invitrogen, #A21203). Both antibodies were diluted in PBS containing 5% BSA. Images were acquired using Leica SP8 confocal microscope, using the appropriate filters, at the PICT-IBiSA Imaging Facility in Orsay. At least 30 cells per sample were imaged and the images were processed with ImageJ by using the MIC-MAQ plugin.

## Colocalization with mitochondria

To determine colocalization events between TOM70-stained mitochondria and PLA/RIBOmap signals (spots), individual cells were segmented using the Cellpose deep learning algorithm [74] on the merged 2D Z maximum projection of mitochondrial and nuclear images. The cyto3 model was employed with a cell diameter parameter set to match the expected cell size. Segmentation and subsequent analyses were performed using the *MIC-MAQ* plugin.

Images were then denoised using a Gaussian filter (radius: 1 pixel), and background subtraction was applied using the "Subtract Background" function in ImageJ, with a rolling ball radius of 30 pixels for mitochondria and 10 pixels for spots, to enhance segmentation quality. Mitochondrial networks were segmented by applying an Isodata threshold to the mitochondria channel. Spot detection was carried out using the 3D Watershed Spot Segmentation tool from the 3D ImageJ *Suite* plugin [75].

To evaluate 3D colocalization between the spots and mitochondrial network, the *JACoP* plugin [76] was used for each cell previously detected. The analysis uses centers-particles object-based method in *JACoP*. A spot was considered colocalized with mitochondria if its centroid was located within the segmented mitochondrial area. Results are reported as the mean percentage of colocalized versus noncolocalized spots per cell across three biological replicates.

## SOX10 expression in tumor samples from the TCGA database

Gene expression profiling and comparative analyses of SOX10 expression between tumor and normal tissues were performed using the Gene Expression Profiling Interactive Analysis 2 (GEPIA2) web server (http://gepia2.cancer-pku.cn/#index). GEPIA2 provides standardized processing and visualization of RNA sequencing data from large-scale cancer and normal tissue datasets. GEPIA2 integrates data from The Cancer Genome Atlas (TCGA) and the Genotype-Tissue Expression (GTEx) project. Specifically, tumor samples were obtained from TCGA while normal samples were obtained from TCGA (adjacent nontumor tissues) and GTEx (healthy tissues), depending on availability. For cross-sample comparison, gene expression values were normalized and presented as transcripts per million (TPM), which accounts for both sequencing depth and gene length, allowing reliable comparison of expression levels across tissues and conditions. All analyses were conducted using GEPIA2's built-in statistical modules.

Overall survival (OS) analysis for SOX10 expression in melanoma was conducted using the Survival Analysis module in GEPIA2. Patients were stratified into expression groups using two cutoff strategies: (1) Median cutoff—Patients were divided into *high-* and *low-*expression groups based on the median SOX10 expression across the TCGA cohort, (2) Quartile cutoff—Patients were divided into *high-expression* (top 25%) and *low-expression* (bottom 25%) groups, providing a stricter contrast between cohorts. For both strategies, Kaplan–Meier survival curves were generated. Statistical significance was assessed using the log-rank test, and the hazard ratio (HR) with 95% confidence interval (CI) was reported. Analyses were performed with GEPIA2 default parameters, with confidence interval display enabled.

Data Accessibility. All RNA-sequencing data used in this study are publicly available (TCGA: https://portal.gdc.cancer.gov and GTEx: https://gtexportal.org/home/). Within GEPIA2, the Data Source section provides details on dataset origin, processing, and normalization. Researchers may also directly access raw and processed data from the TCGA and GTEx repositories in accordance with their respective data access policies.

## Quantification and statistical analysis

Statistical analyses were performed on GraphPad Prism (Version 10.3.0) and statistical significance defined as a *p*-value ≤0.05 was determined by two-tailed unpaired Student *t* test. Comparisons in multiple groups were analyzed with one-way or two-way ANOVA, as indicated in the figure legends. All data in this study are presented as the mean standard deviation or mean standard error.

## Supporting information

**S1 Fig. HK2 expression in melanoma cell lines and tumors. (A)** HK2 expression level assessed by microfluidic western blot analysis of normal melanocytes, early superficial melanoma with radial growth (RGP, SBCL2), early invasive melanoma (VGP, WM793), low invasive (SKMel10) and high invasive (A375) metastatic melanoma cell lines. Actin was used as loading control. Quantification of HK2 protein levels are normalized to actin expression and presented in red, relative to HK2 expression in A375 cells. **(B)** Immunohistochemical analysis of HK2 expression in three representative samples from a cohort of 31 patients with cutaneous melanoma. Left: representative melanoma sample with low HK2 expression. Right: representative melanoma sample with high HK2 expression. **(C)** Western blot analysis of key glycolytic enzymes (HK2, PKM2 and GAPDH) in a variety of melanoma cell lines. Tubulin was used as loading control.
(TIF)

**S2 Fig. Impact of HK2 depletion and glucose inhibition in the metabolic profile of A375 melanoma cells. (A)** (Left panel) western blot analysis of HK2, GAPDH, and PKM2 protein levels in A375 melanoma cells upon siRNA-mediated depletion of HK2 (siHK2, light red) or control (siCTRL, gray). VCL was used as loading control. (Right panel) Quantification of HK2, GAPDH, and PKM2 protein levels are normalized to VCL expression. $p$-values were calculated by ordinary two-way ANOVA with Šídák's multiple comparisons test (SD, $n=3$ biological replicates) and only significant comparisons are shown (** $p \leq 0.01$, *** $p \leq 0.001$). **(B)** Measurement of lactate secretion in A375 cells in A375 melanoma cells upon siRNA-mediated depletion of HK2 (siHK2, light red) or control (siCTRL, gray). $p$-values were calculated by unpaired, two-tailed Student $t$ test test (SD, $n=3$ biological replicates) and only significant comparisons are shown (* $p \leq 0.05$). **(C)** Right panel: the extra-cellular acidification rate (ECAR), an indicator of aerobic glycolysis, was measured in A375 cells grown in either glucose or galactose-containing medium followed by consecutive treatments of glucose (glc), oligomycin, and the HK2 inhibitor 2-DG ($n=1$). Left panel: the oxygen consumption rate (OCR) was measured in A375 cells cultured either in glucose or galactose-containing medium followed by consecutive treatments of oligomycin, FCCP, and antimycin A and rotenone ($n=1$). The individual numerical values for panels S2A–S2C Fig are available at S8 Data.
(TIF)

**S3 Fig. HK2 regulates the expression of cancer-related genes. (A)** RT-qPCR quantification of 84 EMT-associates genes. Transcriptional (total RNA) and translational (polysomal RNA) levels of each gene were obtained from polysome profiling of A375 cells transfected with siRNAs targeting HK2 (siHK2) or control (siCTR) and measured using RT2 Profiler PCR Array ($N=1$). **(B)** Polysome profile of A375 cells upon siRNA-mediated depletion of HK2 (siHK2, light red) or control (siCTRL, gray), assessed by sucrose-gradient (10%–50%) ultracentrifugation. **(C)** western blot analysis of the SOX10 protein level in MALME 3M melanoma cell line transfected with siRNAs targeting HK2 (siHK2, light red), GAPDH (siGAPDH, light gray) or control (siCTRL, gray). The SOX10 protein quantification is normalized to VCL expression. $p$-values were calculated by two-tailed unpaired $t$ test (SD, $n=3$ biological replicates) and only significant comparisons are shown (* $p \leq 0.05$). **(D)** RT-qPCR quantification of the *SOX10* mRNA levels in different melanoma cell lines transfected with siRNAs targeting HK2 (siHK2, light red), GAPDH (siGAPDH, light gray) or control (siCTRL, gray). $p$-values were calculated by ordinary one-way ANOVA (SD, $n=3$ biological replicates). **(E)** RT-qPCR quantification of the *SOX10* mRNA level in A375 cells upon stable HK2 knockdown. $p$-values were calculated by two-tailed unpaired $t$ test (SD, $n=3$ biological replicates). **(F)** western blot quantification of the HK2 protein level in A375 cells upon stable HK2 knockdown. HK2 protein levels are normalized to VCL expression. $p$-value was calculated by two-tailed unpaired $t$ test (SD, $n=3$ biological replicates) (* $p \leq 0.05$). **(G)** RT-qPCR quantification of the *HK2* mRNA level in A375 cells upon stable HK2 knockdown. $p$-values were calculated by two-tailed unpaired $t$ test (SD, $n=3$ biological replicates) (*** $p \leq 0.001$). The individual numerical values for panels S3A–S3G Fig are available at S9 Data.
(TIF)

**S4 Fig. SOX10 expression is associated with prognosis in patients with cutaneous melanoma. (A)** SOX10 expression in different type of solid cancers from TGCA database (https://www.cancer.gov/ccg/research/genome-sequencing/tcga). T: tumor (red dots); N: normal tissue (green dots). **(B)** Analysis of the impact of SOX10 expression in overall survival of patients with cutaneous melanoma. **(C)** Analysis of the impact of SOX10 expression in overall survival according to molecular profile of the tumor. See Materials and methods for information on how the graphs were generated.
(TIF)

**S5 Fig. RNA-binding by HK2 is specific and modulated by RNase treatment. (A)** Western blots of HK2 and GAPDH (normalization control) input of the CLIP from endogenous HK2 in A375 cells presented in Fig 3B. **(B)** CLIP from endogenous HK2 in A375 cells ($n = 3$ biological replicates). Left upper panel: autoradiography of HK2-RNA complexes in UV-C treated (+) A375 cells upon increasing concentrations of RNase I. One major band is observed in the expected molecular mass of HK2, as indicated by a black arrow, and the asterisk (*) indicates a nonspecific band. Left middle and lower panel: western blots of HK2 and IgG immunoprecipitation, and HK2 and GAPDH (normalization control) input, respectively. Same conditions as the upper panel. Right panel: autoradiography of RNA purified from $^{32}$P labeled RNA-HK2 complexes (observed in the left upper panel), migrated on a denaturing TBE-urea gel. The purified RNA migrates as a smear, and it is sensitive to increasing concentrations of RNase I. **(C)** CLIP from endogenous HK2 in A375 cells upon siRNA-mediated depletion of HK2 (siHK2) in comparison to control (siCTRL) ($n = 3$ biological replicates). Upper panel: autoradiography of HK2-RNA complexes in UV-C (+) and RNase I treated (0.05 U/μL) A375 cells. One major band is observed in the expected molecular mass of HK2, as indicated by a black a3rrow. Left middle and lower panel: western blots of HK2 and IgG immunoprecipitation, and HK2 and GAPDH (normalization control) input, respectively. Same conditions as the upper panel. **(D)** CLIP from endogenous HK2 in A2058 cells upon siRNA-mediated depletion of HK2 (siHK2) in comparison to control (siCTRL) ($n = 3$ biological replicates), same conditions as in (C). One major band is observed in the expected molecular mass of HK2, as indicated by a black arrow, and the asterisk (*) indicates a nonspecific band. **(E)** Western blots of GFP input of the CLIP from GFP-only or GFP-HK2 transfected HEK293T cells presented in Fig 3C. The arrow indicates the expected HK2-GFP molecular mass, and the asterisk (*) indicates a nonspecific band. FL: full length; MBD: mitochondrial-binding deficient; DA: nonglucose-binding mutant; SA: catalytically inactive mutant.
(TIF)

**S6 Fig. Sucrose density gradient centrifugation and fractionation of A375 cell lysates.** Western blot quantification representing the % of **(A)** HUR, **(B)** H3, **(C)** GAPDH, and **(D)** PKM2 in each sucrose fraction of lysates treated with RNase I/A/T1 or left untreated (SD, $n = 4$ biological replicates). The individual numerical values for panels S6A–S6D Fig are available at S10 Data.
(TIF)

**S7 Fig. Analysis of HK2-*SOX10* mRNA interaction. (A)** western blot of HK2 and GAPDH (normalization control) from RIP experiment performed on A375 and **(B)** A2058 melanoma cell lines (representative images, $n = 3$ biological replicates). **(C)** *In silico* prediction of HK2-SOX10 mRNA interaction propensities using the *CatRAPID* algorithm. The highest interaction score observed between HK2 and the *SOX10* mRNA is at the *SOX10* 5′UTR. Upper panel: HK2-SOX10 mRNA interaction profile. Lower panel: schematic representation of the *SOX10* mRNA. **(D)** western blot of HK2 and GAPDH (normalization control) from RIP experiment performed on A375 cells transfected with luciferase reporters containing the *SOX10* 5′UTR and 3′UTR sequences upstream of the RLuc luciferase reporter gene. An empty reporter was used as control (representative images, $n = 3$ biological replicates). **(E)** western blot of HK2 and GAPDH (normalization control) from RIP experiment performed on A375 cells transfected with luciferase reporters containing the *SOX10* 5′UTR and the Δ4, Δ5, and Δ6 deletion mutant sequences upstream of the RLuc luciferase reporter gene. An empty reporter was used as

control (representative images, *n* = 3 biological replicates). The individual numerical values for panels S7C Fig are available at S11 Data.
(TIF)

**S8 Fig. SOX10 mRNA levels are not sensitive to glucose or G-6-P concentrations. (A)** RT-qPCR quantification of the *SOX10* mRNA level in A375 cells cultured for 24 h with decreasing concentrations of glucose or under glucose starvation, as indicated (SD, *n* = 3 biological replicates). **(B)** RT-qPCR quantification of the *SOX10* mRNA level in A375 cells cultured in media without glucose supplemented or not with G-6-P (50 mM) for 24 h (SD, *n* = 3 biological replicates). The individual numerical values for panels S8A, S8B Fig are available at S12 Data.
(TIF)

**S9 Fig. RIBOmap reveals *SOX10* mRNA-specific ribosome occupancy. (A)** Representative confocal images of translating *SOX10* mRNAs (green spots) detected by RIBOmap assay in A375 cells transfected with control siRNA (siCTRL) or with siRNA targeting SOX10 (siSOX10). Nuclei are stained with DAPI (blue). Scale bar: 10 μm. **(B)** Quantification of the RIBOmap signal in the condition described in (A). The data shown represent the number of spots/cells from a representative experiment (*n* = 3 biological replicates). *p*-values were calculated by unpaired, two-tailed Student *t* test (**** *p* < 0.0001). **(C)** RT-qPCR quantification of *SOX10* mRNA level in the condition described in g. mean ± SD, *n* = 3 biological replicates. *p*-values were calculated by unpaired, two-tailed Student t *t*est (**** *p* < 0.0001). The individual numerical values for panels S9B Fig are available at S13 Data.
(TIF)

**S10 Fig. HK2 depletion impairs melanoma cell proliferation via the *SOX10* UTR, but affects migration independently. (A)** Scratch-wound assay comparing the migratory capacity of A375 cells upon stable HK2 knockdown (shHK2, light red) to parental cells (shCTRL, gray), ectopically expressing or not SOX10 (ΔUTR SOX10—dashed lines—and Empty, respectively). Upper panel: representative images of the wounded areas of the first acquisition at time zero (T(0)), and at the last acquisition after 72 h (T(72)). Scratch wound area is shown in yellow, initial scratch wound covered by the cells is shown in blue, and the cells adjacent to the initial scratch wound are shown in gray. Lower panel: Relative Wound Density (%) was used to report the data, which was calculated by measuring the spatial cell density in the wounded area relative to the density outside of the same area at the indicated time points. Cells and gap areas were photographed every 3 h for 72 h. Significance was calculated by comparing HK2 knockdown to parental cells after 72 h. *p*-values were calculated by ordinary two-way ANOVA with Turkey's multiple comparison test (SD, *n* = 3 biological replicates), and only significant differences within the same cell line are shown (* *p* ≤ 0.05; ** *p* ≤ 0.01). **(B)** Proliferation assay comparing the percent of plate coverage (cell confluence) of A375 or A2058 cells upon siRNA-mediated depletion of HK2 (shHK2) to parental cells (shCTRL), ectopically expressing or not SOX10 (ΔUTR SOX10 and Empty). Upper panel: representative images of the cell confluence of A375 cells at the moment of the first acquisition at day zero (T(0)), and at the last acquisition after 6 days (T(6)) (*n* = 3 biological replicates, quantifications in the left panel of Fig 7D). Lower panel: representative images of the cell confluence of A2058 cells at the moment of the first acquisition at day zero (T(0)), and at the last acquisition after 6 days (T(6)) (*n* = 3 biological replicates, quantifications in the right panel of Fig 7D). Area not covered by the cells is shown in gray, while covered area is shown in yellow. The individual numerical values for panels S10A Fig are available at S14 Data.
(TIF)

**S11 Fig. PCA plot using the variance stabilizing transformation (VST) data.** The VST offered by DESeq2 was used on the raw count data to stabilize the variance across the mean. The principal components analysis (PCA) plot was built using the ggplot2 package. The individual numerical values for panels S11 Fig are available at S15 Data.
(TIF)

**S1 Table. Translatome data normalized to the expression levels of the corresponding mRNAs in A375 melanoma cells.** Cells were depleted or not of different glycolytic enzymes (i.e., HK2, GAPDH, PKM2). Cells were grown in the presence of galactose instead of glucose were indicated.
(XLSX)

**S2 Table. Transcriptome and translatome analyses on RT² Profiler PCR Array.** Targeted RT-qPCR-based small-scale screen using a panel of a panel of 84 cancer-associated mRNAs.
(DOCX)

**S3 Table. Relevant oligonucleotides.** RT-qPCR primers, EMSA RNA oligonucleotides, PLA and RIBOmap probes.
(DOCX)

**S1 Raw images.** Uncropped blots and gels.
(PDF)

**S1 Data. Individual values for Fig 1.** Individual numerical values for panels Fig 1C and 1F.
(XLSX)

**S2 Data. Individual values for Fig 2.** Individual numerical values for panels Fig 2A–2E, 2G, and 2H.
(XLSX)

**S3 Data. Individual values for Fig 3.** Individual numerical values for panels Fig 3D.
(XLSX)

**S4 Data. Individual values for Fig 4.** Individual numerical values for panels Fig 4A–4D, 4G, 4H, 4J, and 4K.
(XLSX)

**S5 Data. Individual values for Fig 5.** Individual numerical values for panels Fig 5E and 5H.
(XLSX)

**S6 Data. Individual values for Fig 6.** Individual numerical values for panels Fig 6E, 6G, 6I, and 6K.
(XLSX)

**S7 Data. Individual values for Fig 7.** Individual numerical values for panels Fig 7B–7D.
(XLSX)

**S8 Data. Individual values for S2 Fig.** Individual numerical values for panels S2A–S2C Fig.
(XLSX)

**S9 Data. Individual values for S3 Fig.** Individual numerical values for panels S3A–S3G Fig.
(XLSX)

**S10 Data. Individual values for S6 Fig.** Individual numerical values for panels S6A–S6D Fig.
(XLSX)

**S11 Data. Individual values for S7 Fig.** Individual numerical values for panels S7C Fig.
(XLSX)

**S12 Data. Individual values for S8 Fig.** Individual numerical values for panels S8A and S8B Fig.
(XLSX)

**S13 Data. Individual values for S9 Fig.** Individual numerical values for panels S9B Fig.
(XLSX)

**S14 Data. Individual values for S10 Fig.** Individual numerical values for panels S10A Fig.
(XLSX)

**S15 Data. Individual values for S11 Fig.** Individual numerical values for panels S11 Fig.
(XLSX)

## Acknowledgments

Data management, quality control, and primary analysis were performed by the Bioinformatics platform of the Institut Curie. We acknowledge the help of Eva Guerin, Geraldine Gencic, and Drice Challal for the Incucyte, the Seahorse, and the initial CLIP experiments, respectively.

## Author contributions

**Conceptualization:** Ana Luisa Dian, Lucilla Fabbri, Antoine Moya-Plana, Giuseppina Claps, Caroline Robert, Stéphan Vagner.

**Formal analysis:** Ana Luisa Dian, Lucilla Fabbri, Antoine Moya-Plana, Giuseppina Claps, Virginie Quidville, Dorothée Baïlle, Laetitia Besse, Cédric Messaoudi, Wael M. Rabeh, Caroline Robert, Stéphan Vagner.

**Funding acquisition:** Caroline Robert, Stéphan Vagner.

**Investigation:** Ana Luisa Dian, Lucilla Fabbri, Antoine Moya-Plana, Giuseppina Claps, Juliana C. Ferreira, Virginie Quidville, Sylvain Martineau, Dorothée Baïlle, Séverine Roy, Virginie Raynal, Sylvain Baulande.

**Methodology:** Lucilla Fabbri, Céline M. Labbé, Sylvain Martineau.

**Software:** Céline M. Labbé, Laetitia Besse, Cédric Messaoudi.

**Supervision:** Caroline Robert, Stéphan Vagner.

**Writing – original draft:** Ana Luisa Dian, Stéphan Vagner.

**Writing – review & editing:** Ana Luisa Dian, Lucilla Fabbri, Wael M. Rabeh, Caroline Robert, Stéphan Vagner.

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
