## [Editor Report · Decision Letter 0]

28 Jan 2025

Dear Dr Vagner, 

Thank you for submitting via Review Commons your manuscript entitled "An extra-glycolytic function for hexokinase 2 as an RNA-binding protein regulating SOX10 mRNA translation in melanoma" for consideration as a Research Article by PLOS Biology.

Your manuscript has now been evaluated by the PLOS Biology editorial staff as well as by an academic editor with relevant expertise and I am writing to let you know that we would like to invite you to submit a revision in due course.

However, before we can send you the decision with the details, we need you to complete your submission by providing the metadata that is required for full assessment. To this end, please login to Editorial Manager where you will find the paper in the 'Submissions Needing Revisions' folder on your homepage. Please click 'Revise Submission' from the Action Links and complete all additional questions in the submission questionnaire.

Once your full submission is complete, your paper will undergo a series of checks. After your manuscript has passed the checks I will send you the decision. To provide the metadata for your submission, please Login to Editorial Manager (https://www.editorialmanager.com/pbiology) within two working days, i.e. by Jan 30 2025 11:59PM.

Kind regards,

Ines

--

Ines Alvarez-Garcia, PhD

Senior Editor

PLOS Biology

---

## [Editor Report · Decision Letter 1]

29 Jan 2025

Dear Dr Vagner,

Thank you for providing the metadata for your manuscript entitled "An extra-glycolytic function for hexokinase 2 as an RNA-binding protein regulating SOX10 mRNA translation in melanoma" for consideration at PLOS Biology.

As I mentioned, I discussed the manuscript, the Review Commons reports and your revision plan with an Academic Editor expert in the field and we would like to invite you to submit a revision that thoroughly address the reviewers' reports. It will be important to add more information on the other identified targets to validate them and broaden the scope of the manuscript. Regarding the characterisation of SOX10, you should show whether HK2 specifically regulates SOX10 or has a broader role.

Given the extent of revision needed, we cannot make a decision about publication until we have seen the revised manuscript and your response to the reviewers' comments. Your revised manuscript is likely to be sent for further evaluation by all or a subset of the reviewers.

**IMPORTANT - SUBMITTING YOUR REVISION**

3. Resubmission Checklist

a) *PLOS Data Policy*

b) *Published Peer Review*

d) *Blurb*

Please also provide a blurb which (if accepted) will be included in our weekly and monthly Electronic Table of Contents, sent out to readers of PLOS Biology, and may be used to promote your article in social media. The blurb should be about 30-40 words long and is subject to editorial changes. It should, without exaggeration, entice people to read your manuscript. It should not be redundant with the title and should not contain acronyms or abbreviations. For examples, view our author guidelines: https://journals.plos.org/plosbiology/s/revising-your-manuscript#loc-blurb

Sincerely,

Ines

--

Ines Alvarez-Garcia, PhD

Senior Editor

PLOS Biology

---

## [Decision Letter · Decision Letter 2]

14 Jul 2025

Dear Dr Vagner,

Thank you for your patience while we considered your revised manuscript entitled "An extra-glycolytic function for hexokinase 2 as an RNA-binding protein regulating mRNA translation" for publication as a Research Article at PLOS Biology. This revised version of your manuscript has been evaluated by the PLOS Biology editors, the Academic Editor and two of the original reviewers.

Based on the reviews, we are likely to accept this manuscript for publication, provided you satisfactorily address the remaining points raised by Reviewer 2. Please also make sure to address the data and other policy-related requests stated below my signature.

In addition, we would like you to consider a suggestion to improve the title:

"Hexokinase 2 is an RNA-binding protein that regulates mRNA translation and induces melanoma cell proliferation independently of glycolysis"

We expect to receive your revised manuscript within two weeks. 

*Published Peer Review History*

*Press*

Sincerely,

Ines

--

Ines Alvarez-Garcia, PhD

Senior Editor

PLOS Biology

Fig. 1A, C, F; Fig. 2A-E, G, H; Fig. 3D; Fig. 4A-D, G, H, J, K; Fig. 5E, H; Fig. 6E, G, I, K; Fig. 7B, C, D; Fig. S2A-C; Fig. S3A-G; Fig. S4A-C; Fig. S6A-D; Fig. S7C; Fig. S8A, B; Fig. S9B, C; Fig. S10A and Fig. S11

**In addition, please make publicly available the data you have deposited at the GEO database (ID: GSE274146).

CODE POLICY

We require the original, uncropped and minimally adjusted images supporting all blot and gel results reported in an article's figures or Supporting Information files. We will require these files before a manuscript can be accepted so please prepare and upload them now. Please carefully read our guidelines for how to prepare and upload this data: https://journals.plos.org/plosbiology/s/figures#loc-blot-and-gel-reporting-requirements

We would need the original images for the gels shown in the following figures:

Fig. 1A, B; Fig. 1B-E; Fig. 3A-D; Fig. 5C, D, G; Fig. 6A, B; Fig. 7A, B, Fig. S1A, C; Fig. S2A; Fig. S3C; Fig. S5A-E and Fig. S7A, B, D, E

Reviewers' comments

Rev. 2: Laura Broglia - note that this reviewer has signed the review

General Evaluation

This manuscript explores a novel, non-metabolic role of Hexokinase 2 (HK2) as an RNA-binding protein (RBP) that regulates mRNA translation in melanoma cells. While the metabolic functions of glycolytic enzymes in cancer are well established, their emerging roles in translational control are less well defined. The authors present compelling evidence that HK2 directly binds to the 5′ UTR of SOX10 mRNA and promotes its translation independently of its enzymatic activity and metabolic function. These findings highlight a previously unrecognized function of HK2 in tumor cell proliferation and offer valuable insights into the interplay between metabolism and gene expression regulation in cancer.

The authors have addressed all my major concerns raised in the first round of revision. The manuscript is now much improved, with the inclusion of key experiments that convincingly support the role of HK2 as a regulator of SOX10 mRNA translation.

Comments

- While I understand the rationale for focusing on the selected targets, the purpose of the translatome and RNA-seq analyses remains somewhat unclear and not connected with the rest of the paper. Since the six targets of interest were ultimately identified by another strategy, it would be helpful to specify whether they were absent from the RNA-seq data due to a lack of statistical significance. If so, please indicate in page 12, line 9 the p-values for those genes to show how close they were to the significance threshold (e.g., p < 0.05).

- Figure 1F: For the qPCR results, please avoid setting the control arbitrarily to 1. Use proper normalization, as done for the Western blot quantification.

- Figure 2A: Please include statistical analysis for the different fractions.

- Figure 6E: Statistical information is missing in the plot. Please include it for completeness.

- Regarding the rescue experiment: I originally suggested including ectopic expression of SOX10 ORF with both the WT 5′ UTR and the 5′ UTR harboring deletion #4 (i.e., the deletion that showed reduced luciferase activity and lower enrichment in IP).

The authors respond that this experiment is not feasible due to inefficient translation of the transduced ORF with the deleted 5′ UTR. My original point was that such a control would be critical to demonstrate that the deleted region is indeed necessary for regulation (i.e. no rescue should be observed when using that construct). That said, I acknowledge that this conclusion is already well supported by the other complementary approaches.

Minor Comments

- Abstract, lines 25–26: The sentence "by promoting the release of HK2..." is unclear. Please clarify—do high glucose conditions promote HK2 release from the outer mitochondrial membrane (OMM)? Rephrasing is needed for clarity.

- Introduction, page 7, line 31: Please define or briefly explain SNAIL-mediated EMT for clarity.

- Introduction, page 8, line 6: Spell out and define "OXPHOS" at first mention.

- Page 14, lines 33–35: Rephrase this sentence. It is currently unclear and appears to misuse “either…or.”

- Page 28, lines 16–18: Add a reference to the relevant figure.

Rev. 3:

Accept.

---

## [Editor Report · Decision Letter 3]

12 Aug 2025

Dear Dr Vagner,

Thank you for the submission of your revised Research Article entitled "Hexokinase 2 is an RNA-binding protein that regulates mRNA translation independently of glycolysis and induces melanoma cell proliferation" for publication in PLOS Biology. On behalf of my colleagues and the Academic Editor, Elena Rainero, I am delighted to let you know that we can in principle accept your manuscript for publication, provided you address any remaining formatting and reporting issues. These will be detailed in an email you should receive within 2-3 business days from our colleagues in the journal operations team; no action is required from you until then. Please note that we will not be able to formally accept your manuscript and schedule it for publication until you have completed any requested changes.

PRESS

Sincerely, 

Ines

--

Ines Alvarez-Garcia, PhD

Senior Editor

PLOS Biology
